# DOCS: Quantifying Weight Similarity for Deeper Insights into Large Language Models

**Zeping Min**
Alibaba Group
Hupan Laboratory
AMSS, Chinese Academy of Sciences
minzeping.mzp@alibaba-inc.com

**Xinshang Wang**
Alibaba Group
xinshang.w@alibaba-inc.com

## Abstract

We introduce a novel index, the Distribution of Cosine Similarity (DOCS), for quantitatively assessing the similarity between weight matrices in Large Language Models (LLMs), aiming to facilitate the analysis of their complex architectures. Leveraging DOCS, our analysis uncovers intriguing patterns in the latest open-source LLMs: adjacent layers frequently exhibit high weight similarity and tend to form clusters, suggesting depth-wise functional specialization. Additionally, we prove that DOCS is theoretically effective in quantifying similarity for orthogonal matrices, a crucial aspect given the prevalence of orthogonal initializations in LLMs. This research contributes to a deeper understanding of LLM architecture and behavior, offering tools with potential implications for developing more efficient and interpretable models.

## 1 Introduction

Large Language Models (LLMs), built on transformer architectures (Vaswani et al., 2017), have ushered in a new era in natural language processing (Brown et al., 2020). These complex models have demonstrated remarkable capabilities, but understanding their underlying mechanisms remains a challenge. Similarity analysis techniques (Raghu et al., 2017; Morcos et al., 2018; Kornblith et al., 2019) offer a promising approach for gaining insights into the learned representations and computational processes within these models.

In this work, we extend the application of similarity analysis by directly examining the *weight matrices* of various LLMs[1], instead of focusing on representations. By analyzing the weights themselves, we aim to uncover deeper insights into the model's structure and functionality that are not apparent from representations alone.

While prior research has explored many methods for characterizing the similarity of neural networks (Wu et al., 2020; Khosla & Williams, 2024; Kriegeskorte et al., 2008; Klabunde et al., 2023b; Wang et al., 2020; Barannikov et al., 2021; Hamilton et al., 2016; Rahamim & Belinkov, 2024; Tang et al., 2020; Camastra & Staiano, 2016; Wang et al., 2018; Raghu et al., 2017; Morcos et al., 2018; Kornblith et al., 2019), these methods are often not applicable to measuring similarity between weight matrices due to the following two key factors. For further discussion, see Appendix E.

1. **Focus on Representation, Not Weights:** Similar representations across layers do not necessarily imply similar weight matrices. This discrepancy arises from the use of *residual connections* in transformer architectures (He et al., 2016), which create shortcuts that allow information to bypass layer transformations. Mathematically, a residual connection is represented as

$$\mathbf{y} = \mathcal{F}(\mathbf{x}, \mathcal{W}) + \mathbf{x}, \tag{1}$$

where $\mathbf{x}$ is the layer's input, $\mathcal{W}$ represents the weight matrices, $\mathcal{F}$ is the transformation function (including the feedforward network and attention), and $\mathbf{y}$ is the layer's output. As

---

[1]While different transformer-based architectures exist, in this paper we primarily focus on *decoder-only* architectures, which include the $W_v$, $W_k$, $W_q$, $W_o$, MLP-Up, and MLP-Down weight matrices.

shown in equation 1, the output $\mathbf{y}$ directly depends on the input $\mathbf{x}$. While residual connections mitigate issues such as vanishing gradients during training, this direct dependence inherently creates *correlations* between the inputs of different layers, as the output of one layer serves as the input for the next. Since the transformation function $\mathcal{F}$ depends on the input $\mathbf{x}$, these input correlations can lead to *correlated representations* of transformations across layers, even if the underlying weight matrices $\mathcal{W}$ are distinct. This is further evidenced by Figures 1a and 1b, which show that the input and output of the feedforward network have similar patterns of representation similarity.

Consequently, observing similar representations across layers does not guarantee that the corresponding weight matrices are also similar. Since representation and weight are two fundamental facets of the model, each offers unique insights. Therefore, while representation analysis can provide profound understanding of large language models, examining the weight matrices can further deepen our comprehension of their structure and behavior, offering additional perspectives for potential applications. For further discussion, see Appendix E.1.

2. **Non-Discriminative for Orthogonal Matrices:** Many existing similarity indices, such as Canonical Correlation Analysis (CCA) (Ramsay et al., 1984; Morcos et al., 2018), Singular Vector Canonical Correlation Analysis (SVCCA) (Raghu et al., 2017), and linear Centered Kernel Alignment (linear CKA) (Kornblith et al., 2019), are *non-discriminative for orthogonal matrices*. An *orthogonal matrix* $Q$ is defined by the property $Q^\top Q = QQ^\top = I$, where $I$ is the identity matrix. This non-discriminative nature means that these indices can yield the same score when assessing the similarity between *any two* orthogonal matrices, regardless of their actual differences. This limitation hinders accurate similarity assessment, as it fails to capture genuine differences between weight matrices. This issue is particularly relevant in the context of LLMs, where orthogonal matrices commonly occur throughout the training process (Tian et al., 2023).

To address these challenges, we introduce a novel matrix-similarity index called the *Distribution of Cosine Similarity (DOCS)*. DOCS directly measures the similarity of weight matrices by computing the cosine similarity between corresponding vectors and analyzing their distribution. Importantly, DOCS retains the desirable properties of existing similarity indices while overcoming their non-discriminative nature for orthogonal matrices, a critical factor in LLM analysis. Through extensive experiments on various LLMs, we demonstrate that DOCS provides a more reliable and accurate measure of similarity between LLM weight matrices.

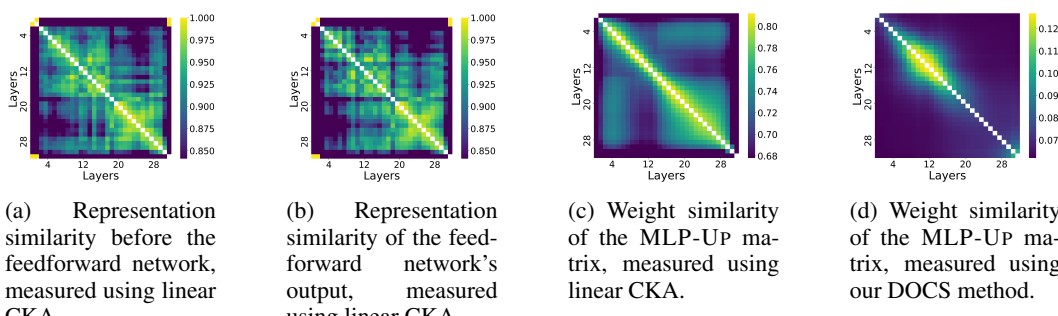

(a) Representation similarity before the feedforward network, measured using linear CKA.

(b) Representation similarity of the feedforward network's output, measured using linear CKA.

(c) Weight similarity of the MLP-Up matrix, measured using linear CKA.

(d) Weight similarity of the MLP-Up matrix, measured using our DOCS method.

Figure 1: Comparison of similarity indices applied to representation similarities ((a) and (b)) and weight similarities ((c) and (d)) across different layers of Llama 3.1-8B-Instruct (Dubey et al., 2024).

To demonstrate the effectiveness of our DOCS method, Figures 1c and 1d compare weight similarity heatmaps generated using linear CKA (Kornblith et al., 2019) and DOCS. The linear CKA heatmap shows light off-diagonal areas and dark stripes, indicating an inability to distinguish between distant layers—possibly due to the orthogonality of weight matrices (see Section 2). In contrast, the DOCS heatmap displays clear light areas near the diagonal, highlighting similarities between adjacent layers in the LLM.

Our results reveal many intriguing similarity patterns within open-source LLMs, leading us to explore several key observations in the following discussions.

**Neighboring Transformer Layers Exhibit Similar Weights.**

Our analysis reveals a consistent pattern of weight similarity between adjacent layers in open-source LLMs. This suggests that after optimization, similar neurons tend to stay in layers that best suit their function. This observation supports recent studies (Lad et al., 2024; Song et al., 2024; Mu et al., 2024) indicating functional redundancy in adjacent layers. These studies have demonstrated high prediction accuracy even after layer manipulations (Lad et al., 2024), identified block-level redundancy (Song et al., 2024), and observed analogous attention patterns in nearby layers (Mu et al., 2024). Furthermore, we find that weight similarity decreases with increasing layer distance, aligning with the hypothesized universal stages of inference across models (Lad et al., 2024).

**Clusters of Similar Transformer Layers Exist.**

Beyond merely adjacent layers, we find that clusters of multiple similar, nearby layers exist within LLMs. As depicted in Figure 1d, layers 7–12 form such a cluster, exhibiting relatively high mutual similarity (approximately twice the DOCS index values of other elements) in their MLP-UP module weights according to our DOCS index. This challenges the common practice of uniform layer configurations, as such designs fail to leverage the cluster structure revealed by DOCS, potentially limiting optimization during the SFT stage.

Many open-source LLM implementations, including GPT-2 (Radford et al., 2019), Llama (Touvron et al., 2023), Mistral (Jiang et al., 2023), Llama 3 (Dubey et al., 2024), Gpt-neox-20b (Black et al., 2022), Opt (Zhang et al., 2022), Codegeex (Zheng et al., 2023), Glm-130b (Zeng et al., 2022), and Flm (Li et al., 2023), adopt architectures where all layers have the same size. Furthermore, existing literature on scaling laws for neural language models (Kaplan et al., 2020; Hoffmann et al., 2022) and parameter-efficient fine-tuning (PEFT) methods (Hu et al., 2021; Houlsby et al., 2019) often assumes uniform layer sizes, treating the model as a homogeneously scaled entity.

Our observations suggest revisiting these assumptions. The presence of mutual similarity within the layers of a cluster indicates the potential to apply a distinct configuration to those layers, such as adjusting neuron sizes or training strategies. DOCS similarity can guide such designs, aligning with prior efforts to leverage layer clusters for reduced computation (Liao & Vargas, 2024).

**Comparing Weight Similarities Between Base and Instruction-Tuned Models.**

We investigate whether base models and their instruction-tuned counterparts exhibit similar weight patterns. We address questions such as: How different are the base and instruction fine-tuned models? Which parts of the LLMs are most affected by instruction fine-tuning? Are there any patterns in the changes of weight matrices due to fine-tuning? This examination sheds light on how instruction tuning affects the internal weights of models (Ouyang et al., 2022).

**Comparing Similarities Between Experts**

We examine the similarity between experts in pre-trained Mixture of Experts (MoE) models. Do the experts have similar weights? Is there any expert that is significantly different from the others? Our analysis provides insights into the diversity among experts (Shazeer et al., 2017; Lepikhin et al., 2020).

## 2 MATHEMATICAL PROPERTIES OF SIMILARITY INDICES

When evaluating similarity indices for neural network weights, especially in the context of large language models, a critical property to consider is their ability to *discriminate between orthogonal matrices.* Orthogonal matrices play a significant role in neural network initialization and training. They are often used to improve training stability and preserve gradient flow. Even after optimization, weight matrices may retain orthogonality properties (Tian et al., 2023). Therefore, a similarity index that can distinguish between different orthogonal matrices is essential for capturing meaningful variations in weight matrices.

We categorize the behavior of similarity indices concerning orthogonal matrices into three classes:

**Definition 1** (Constant on Orthogonal Matrices). *An index $S$ is* constant on orthogonal matrices *if there exists a constant $C \in \mathbb{R}$ such that for all $n \in \mathbb{N}$ and all orthogonal matrices $X, Y \in \mathbb{R}^{n \times n}$, we have $S(X, Y) = C$.*

**Definition 2** (Dimension-Dependent on Orthogonal Matrices). *An index $S$ is* dimension-dependent *on orthogonal matrices if for each $n \in \mathbb{N}$, there exists a constant $C(n) \in \mathbb{R}$ such that for all orthogonal matrices $X, Y \in \mathbb{R}^{n \times n}$, we have $S(X, Y) = C(n)$.*

**Definition 3** (Discriminative on Orthogonal Matrices). *An index $S$ is* discriminative *on orthogonal matrices if there exist orthogonal matrices $X, Y, X', Y' \in \mathbb{R}^{n \times n}$ such that $S(X, Y) \neq S(X', Y')$.*

Apart from the behavior concerning orthogonal matrices, there are other desirable mathematical properties that ensure similarity indices provide meaningful and consistent comparisons across different models and layers. These properties include:

1. **Permutation Transformation (PT) Invariance (Williams et al., 2021)**:

$$S(X, Y) = S(XP_X, YP_Y)$$

   where $P_X$ and $P_Y$ are permutation matrices. This property characterizes the ability of an index to be unaffected by permutations in the neuron ordering.

2. **Symmetry.** $S(X, Y) = S(Y, X)$. The similarity measure should be independent of the order of the inputs, ensuring a fair comparison between two matrices.

3. **Isotropic Scaling (IS) Invariance. (Klabunde et al., 2023a;b)** $S(aX, bY) = S(X, Y)$ for any non-zero scalars $a$, $b$. This allows for meaningful comparisons of models trained under different conditions, such as varying learning rates or initialization schemes that scale the weights differently.

4. **Reflexivity.** $S(X, X) = 1$. A matrix should be most similar to itself, providing a normalization baseline for similarity measures.

Table 1: Comparison of Mathematical Properties Across Different Similarity Indices.

| Method | PT Invariance | Symmetry | IS Invariance | Reflexivity | Behavior on Orthogonal Matrices |
|---|---|---|---|---|---|
| Linear Regression | ✓ | ✗ | ✓ | ✓ | Constant |
| CCA ($R^2_{\text{CCA}}$) (Morcos et al., 2018) | ✓ | ✓ | ✓ | ✗ | Constant |
| CCA ($\bar{\rho}_{\text{CCA}}$) (Morcos et al., 2018) | ✓ | ✓ | ✓ | ✓ | Constant |
| SVCCA ($R^2_{\text{SVCCA}}$) (Raghu et al., 2017) | ✓ | ✓ | ✓ | ✓ | Constant (assuming $T_X = T_Y = I$) |
| SVCCA ($\bar{\rho}_{\text{SVCCA}}$) (Raghu et al., 2017) | ✓ | ✓ | ✓ | ✓ | Constant (assuming $T_X = T_Y = I$) |
| Linear HSIC (Gretton et al., 2005) | ✓ | ✗ | ✗ | ✗ | Dimension-Dependent |
| Linear CKA (Kornblith et al., 2019) | ✓ | ✓ | ✓ | ✓ | Constant |
| DOCS (Ours) | ✓ | ✓ | ✓ | ✓ | **Discriminative** |

Table 1 compares the behavior of various similarity indices with respect to these mathematical properties. Our proposed DOCS index introduces a *discriminative* behavior on orthogonal matrices (see Section 3.1 for theoretical results) and satisfies all the other mathematical properties outlined above. This discriminative capability allows DOCS to capture meaningful differences between weight matrices that other similarity indices—which are constant or dimension-dependent on orthogonal matrices (see proofs in Appendix B) —might overlook. Consequently, DOCS provides a more reliable and accurate measure of similarity between LLM weight matrices, enhancing the analysis and understanding of neural network behaviors.

In Appendix D, we discuss additional mathematical properties, some of which DOCS may not satisfy. These properties, while valuable in other contexts, are not critical for evaluating weight similarity measures in neural networks.

## 3 DISTRIBUTION OF COSINE SIMILARITY

Our DOCS method operates by comparing the weight matrices of two components (e.g., feed-forward networks, attention heads) within a neural network. Each component is represented by a matrix where columns correspond to individual parameter vectors (e.g., neuron weights, attention patterns). The key idea is to quantify how well the parameters from one component align with those from another, based on their vector representations.

---

**Algorithm 1** Computation of the DOCS Similarity Index $S_{\text{DOCS}}$

---

1: **Input:** Matrices $X = [X_1, X_2, \ldots, X_m] \in \mathbb{R}^{n \times m}$ and $Y = [Y_1, Y_2, \ldots, Y_m] \in \mathbb{R}^{n \times m}$
2: **Output:** Similarity index $S_{\text{DOCS}}$

3: **function** MAXCOSSIM($A$, $B$)
4:     Compute the cosine similarity matrix $C \in \mathbb{R}^{m \times m}$ where $C_{jk} = \dfrac{A_j^\top B_k}{\|A_j\|\|B_k\|}$
5:     For each column $A_j$, find $s_{A_j} = \max_k |C_{jk}|$
6:     **return** $\mathbf{s}_A = [s_{A_1}, s_{A_2}, \ldots, s_{A_m}]^\top$
7: **end function**

8: Compute $\mathbf{s}_X = $ MAXCOSSIM($X, Y$)
9: Compute $\mathbf{s}_Y = $ MAXCOSSIM($Y, X$)
10: Fit a Gumbel distribution to $\mathbf{s}_X$ to estimate the location parameter $u_X$ using maximum likelihood estimation
11: Fit a Gumbel distribution to $\mathbf{s}_Y$ to estimate the location parameter $u_Y$ using maximum likelihood estimation
12: Compute the similarity index:
$$S_{\text{DOCS}} = \frac{u_X + u_Y}{2}$$

---

The DOCS algorithm consists of the following steps:

1. **Compute Cosine Similarities:** Calculate the cosine similarity between all pairs of parameter vectors from the two weight matrices. This results in a cosine similarity matrix $C$, where each element $C_{jk}$ quantifies the similarity between the j-th vector from $X$ and the k-th vector from $Y$.

2. **Extract Maximum Similarities:** For each vector in $X$, identify the vector in $Y$ with which it has the highest absolute cosine similarity. This captures the strongest alignment for each vector.

3. **Fit Gumbel Distribution:** Fit separate Gumbel distributions to $\mathbf{s}_X$ and $\mathbf{s}_Y$ by treating the elements in each vector as data points. Using maximum likelihood estimation, compute the location parameters $u_X$ and $u_Y$, respectively. This allows us to summarize each distribution with a single parameter.

4. **Compute DOCS Index:** The location parameters $u_X$ and $u_Y$ of the fitted Gumbel distributions represent the central tendency of the maximum similarities. Averaging $u_X$ and $u_Y$ yields the DOCS similarity index $S_{\text{DOCS}}$.

The DOCS index $S_{\text{DOCS}}$ provides a scalar value between 0 and 1 that reflects the degree of similarity between the two weight matrices. A higher value indicates that the matrices have parameters (e.g., neuron weights or attention patterns) that are highly aligned, suggesting similar functional roles. Conversely, a lower value implies less similarity, indicating that the components may be specialized for different functions.

By focusing on maximum cosine similarities and modeling their distribution, DOCS captures significant parameter alignments instead of averaging over all pairwise similarities. In contrast, similarity indices such as Canonical Correlation Analysis (CCA) (Morcos et al., 2018), Singular Vector CCA (SVCCA) (Raghu et al., 2017), and linear Centered Kernel Alignment (linear CKA) (Kornblith et al., 2019) rely on matrix multiplication to aggregate pair-wise information across entire matrices. This aggregation can dilute strong correspondences between specific parameter vectors, potentially overlooking meaningful alignments. Consequently, DOCS effectively detects strong correspondences between components, enhancing the analysis of deep neural network structures.

### 3.1 THEORETICAL JUSTIFICATION

We establish that DOCS can distinguish between orthogonal matrices, a capability that existing similarity indices lack (see Section 2). The following theorem demonstrates that DOCS not only

meets the Definition 3 of being discriminative on orthogonal matrices but also achieves a *stronger* level of distinction through a constructive proof.

**Theorem 1.** *For $n \geq 2$, there exist $m = \Omega(n)$ and column-orthogonal matrices $X, Y \in \mathbb{R}^{n \times m}$ such that their Frobenius norm difference and DOCS similarity satisfy:*

$$\|X - Y\|_F = \Omega(\sqrt{m}), \quad and \quad S_{DOCS}(X, Y) = \frac{1}{\sqrt{m}}.$$

The proof of the theorem is deferred to Appendix C. The intuition behind the proof is to construct the matrices $X$ and $Y$ with orthonormal columns but differing structures to highlight their dissimilarity. The matrix $X$ is built from standard basis vectors, while $Y$ leverages a normalized Hadamard matrix to ensure orthonormality. By calculating the Frobenius norm of the difference $X - Y$, we show that it scales as $\sqrt{m}$. Meanwhile, the DOCS value between $X$ and $Y$ is controlled by the normalized entries of the Hadamard matrix, demonstrating that their cosine similarity is small, on the order of $1/\sqrt{m}$. This gap between the large Frobenius norm and small DOCS establishes the desired properties.

This theorem proves the existence of column-orthogonal matrices with significant differences $\|X - Y\|_F = \Omega(\sqrt{m})$. Unlike existing methods, DOCS similarity effectively captures these differences $S_{\text{DOCS}}(X, Y) = \frac{1}{\sqrt{m}}$, demonstrating its superior discriminative power for orthogonal matrices (see Table 1). An illustrative example is provided in Appendix A.5.

## 4 EXPERIMENTS

We conduct experiments to demonstrate the capabilities of DOCS and to gain insights into the internal structure of LLMs. In LLM implementations[2], the rows of a weight matrix correspond to output dimensions, and the columns correspond to input dimensions. To align the column vectors with meaningful entities (e.g., neuron weights), we transpose $W_v$, $W_k$, $W_q$, and MLP-UP before computing DOCS scores. In the MoE experiment, $W_1$ and $W_3$ (Jiang et al., 2024) are also transposed for consistency.

### 4.1 COMPARISON OF SIMILARITY INDICES

Figure 2 provides a visual comparison of eight different similarity indices applied to the MLP-UP layers of the Meta-Llama-3.1-8B-Instruct model.

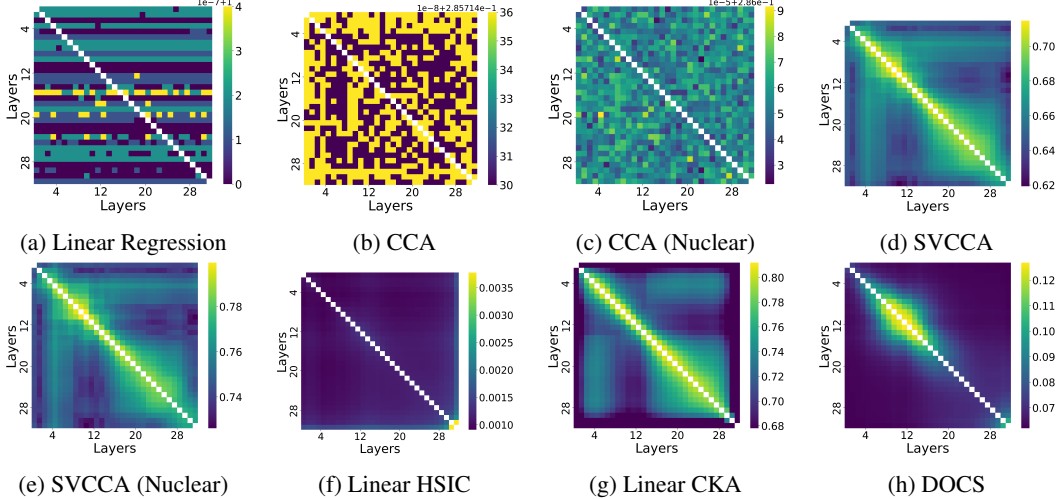

Figure 2: Comparison of similarity indices on the MLP-UP layers of Meta-Llama-3.1-8B-Instruct.

---

[2]https://github.com/huggingface/transformers

In these visualizations, we observe that Linear Regression, Canonical Correlation Analysis (CCA), and CCA (Nuclear) indices fail to exhibit clear structural patterns, with their heatmaps appearing noisy. This suggests that they may have limitations in accurately capturing meaningful relationships between the parameters of different transformer layers. The observed noise could be indicative of their reduced sensitivity to the underlying layer similarities.

On the other hand, the similarity indices Singular Vector CCA (SVCCA), SVCCA (Nuclear), and Linear Centered Kernel Alignment (Linear CKA) display more discernible patterns. Specifically, they exhibit block-diagonal structures, which suggests that the layers in close proximity share higher similarity. This behavior aligns with our expectations, as neighboring layers within transformer models often exhibit greater parameter correlation due to their sequential processing of information. However, we also observe the presence of minor blocks and stripes in the off-diagonal regions, which could be attributed to biases or noise in the similarity indices. These artifacts may result from the indices' reduced ability to differentiate orthogonal matrices, as elaborated in Section 2.

In contrast, the heatmap for DOCS exhibits a clear structure along the diagonal, demonstrating its effectiveness in identifying similar layers with minimal interference from noisy or unrelated signals. Furthermore, we calculated the Gini coefficients of the similarity heatmaps obtained through each method (calculation details are provided in Appendix H). A higher Gini coefficient indicates a more uneven distribution of similarity scores, signifying a greater concentration of significant similarities in fewer layer pairs. This concentration potentially highlights structural characteristics of the model parameters. As shown in Table 2, DOCS achieves the highest Gini coefficient among all the evaluated similarity indices. This result suggests that DOCS excels in revealing structural characteristics of the model parameters.

Table 2: Gini Coefficients for Different Similarity Indices on MLP-Up of Meta-Llama-3.1-8B-Instruct

| Similarity Index | Gini Value |
|---|---|
| CCA Nuclear Similarity | 0.0098 |
| CCA Similarity | 0.0098 |
| Linear CKA Similarity | 0.0488 |
| Linear HSIC Similarity | 0.0617 |
| Linear Regression Similarity | 0.0098 |
| SVCCA Nuclear Similarity | 0.0186 |
| SVCCA Similarity | 0.0225 |
| DOCS (Ours) | **0.0745** |

Appendix E.3 provides more experimental results.

## 4.2 NEIGHBORING LAYERS EXHIBIT SIMILAR WEIGHTS

We investigated the similarity patterns between neighboring transformer layers by analyzing various weight matrices ($W_v$, $W_k$, $W_q$, $W_o$, MLP-Up, MLP-Down) in various LLMs. We employed DOCS to compute and visualize these similarities. Figure 3 illustrates the results for $W_k$, $W_q$, and MLP-Down on gemma-2-27b-it.

The heatmaps clearly show that adjacent layers exhibit higher similarity scores. Scatter plots in Figures (d), (e), and (f) further support this, displaying a decreasing trend in similarity as the absolute layer index difference increases. The shaded areas, representing one standard deviation around the mean similarity, reinforce this observation.

Interestingly, the first and last layers—the most distant ones—also display higher similarity, as indicated by upward trends at the ends of the scatter plots. We hypothesize that this is because both layers are closely connected to the token embeddings. Similar patterns were observed for other weight matrices and across different LLMs.

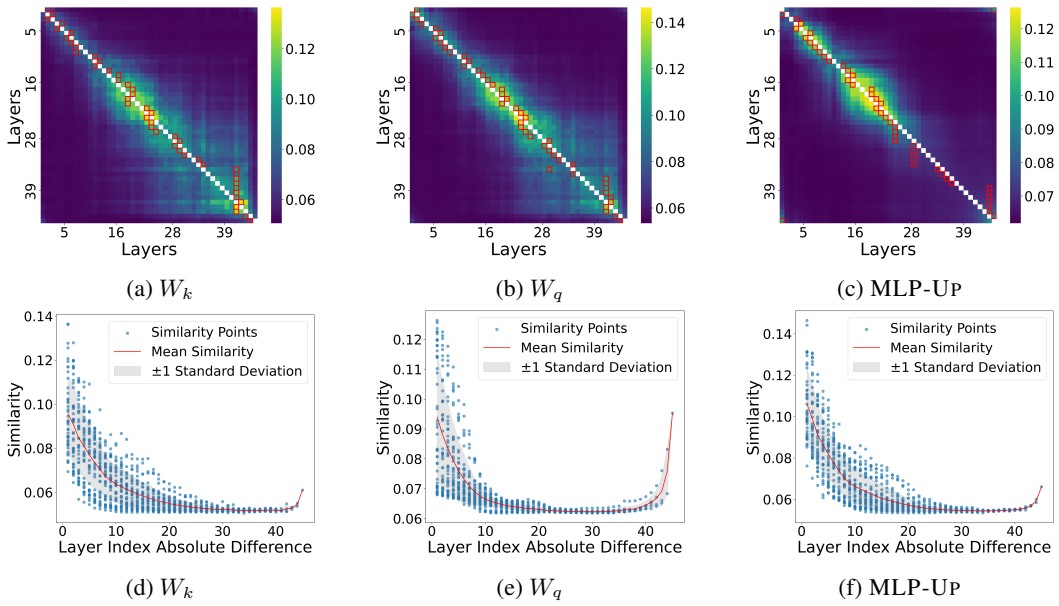

Figure 3: The top row displays heatmaps of DOCS scores between layers for different weight matrices in gemma-2-27b-it. The bottom row illustrates the relationship between DOCS scores and the distance between layers.

Due to space limitations, other results are presented in Appendix I. These findings suggest that neighboring layers tend to share more functional similarities compared to layers that are farther apart.

### 4.3 CLUSTERS OF SIMILAR LAYERS

We examined clusters of similar consecutive layers within LLMs. Figure 4 shows heatmaps of DOCS scores for the $W_v$ matrices. They reveal clusters represented by light regions—each containing multiple layers with high mutual DOCS scores. The bottom row displays the average DOCS scores for diagonal blocks ranging from $3 \times 3$ to $7 \times 7$, providing a clearer view of cluster locations and their similarity levels.

An interesting pattern emerges: each figure shows two clusters of layers. The first cluster, located in the middle depths (centered around layer 19 in gemma-2-9b and gemma-2-27b, and around layer 10 in Llama-3.1-8B and Mixtral-8x7B), is the most distinct. The second cluster appears in the last layers (after layer 27 in gemma-2-9b, after 33 in gemma-2-27b, after 21 in Llama-3.1-8B, and after 23 in Mixtral-8x7B), though it is less pronounced in some models. This phenomenon, observed across different model sizes and even vendors, suggests a universal structural pattern resulting from LLM training. The consistent appearance of these clusters indicates that the training process leads to the formation of distinct functional groups of layers within the model.

Different models, however, may exhibit a varying number of clusters. For example, Figure 5 shows that Llama-3.1-70B contains multiple clusters.

### 4.4 BASE VS. INSTRUCT MODELS

We used the DOCS index to measure changes in weight matrices between base and instruction-tuned models, examining LLM families including yi-1.5, Llama-3.1, and gemma-2. Figure 6 illustrates the results. Our analysis reveals that all DOCS scores are notably high, with values exceeding $0.7$ for every matrix evaluated. This indicates that the base and instruction-tuned models largely retain the same foundational knowledge after the fine-tuning process.

On the other hand, a notable observation from our results is the tendency of the weight matrices to cluster into three distinct groups based on the trends observed in their DOCS scores. Specifically,

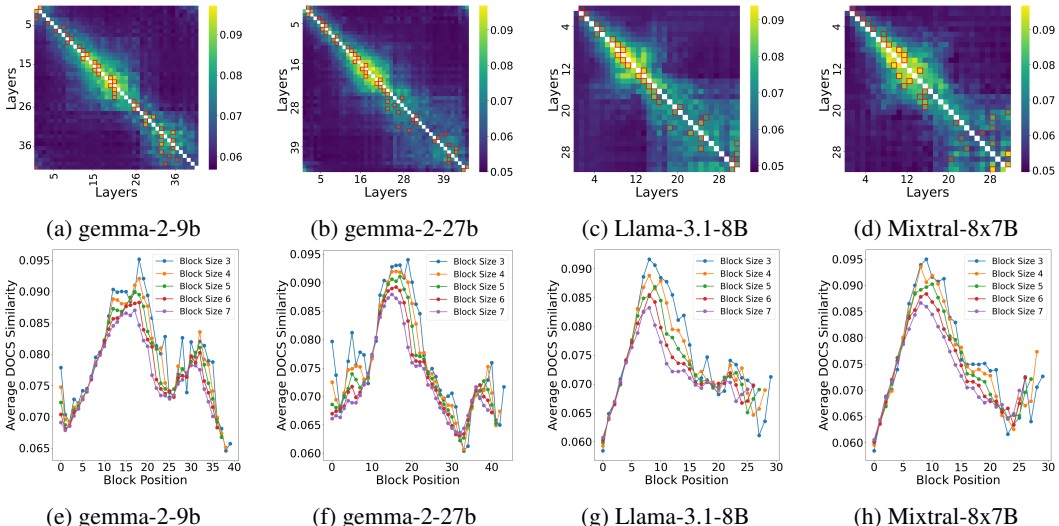

Figure 4: Analysis of $W_v$ matrices across various LLMs. Top row: Heatmaps visualize DOCS similarity scores between transformer layers. Bottom row: Average DOCS scores are computed for diagonal blocks (sizes 3x3 to 7x7) within each heatmap.

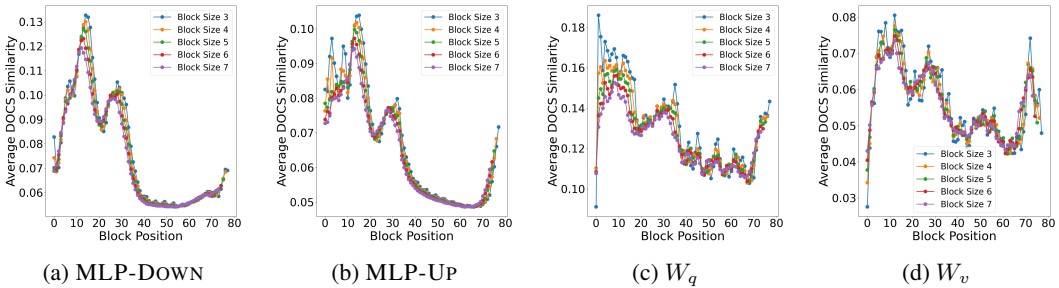

Figure 5: Average DOCS scores for diagonal blocks of varying sizes (3x3 to 7x7) within heatmaps representing different weight matrices in Llama-3.1-70B.

these groups consist of MLP-UP and MLP-DOWN, $W_q$ and $W_k$, as well as $W_v$ and $W_o$. This grouping may be due to the functional similarities between these matrices in the model architecture.

### 4.5 MOE EXPERIMENT

We analyzed the weights of different experts in Mixtral-8x7B using DOCS to generate similarity heatmaps for the expert weight matrices ($W_1$, $W_2$, $W_3$) in the MoE network. The outcomes are visualized in Figure 7.

In many model layers, typically just one expert stands out, indicated by dark grids that show clear separation from the others. Figures 7a, 7b and 7c illustrate this with the third expert's prominent dark intersecting lines, highlighting its uniqueness. This suggests some experts may have specialized roles, differing in function or behavior within the architecture. We hypothesize that this may be related to data load imbalance during the MoE training process, where a large portion of the data concentrates on a single expert (Dai et al., 2022; Zuo et al., 2021).

### 5 CONCLUSION AND FUTURE WORK

This work introduces DOCS, a novel index for quantifying weight similarity in large language models. Unlike existing similarity indices, DOCS effectively differentiates orthogonal matrices, addressing a key limitation and enabling deeper insights into the internal structures of LLMs. Our

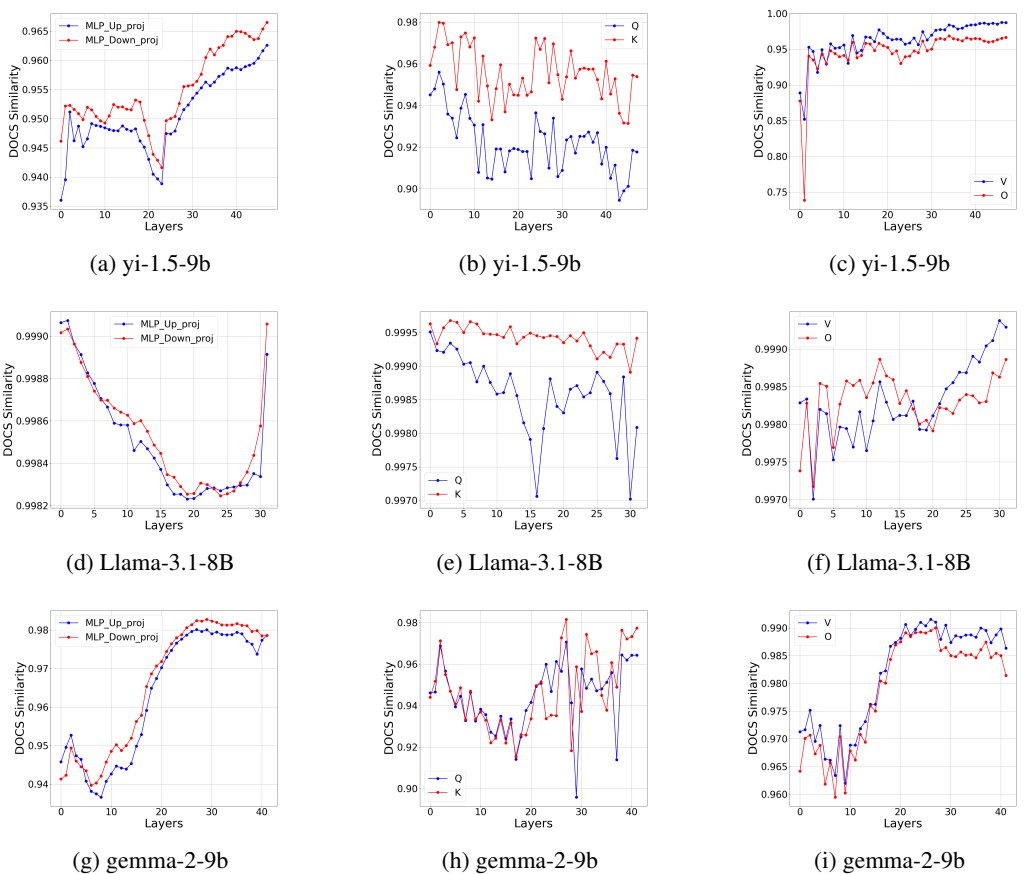

Figure 6: DOCS similarity scores between the base and instruction fine-tuned weight matrices for various models.

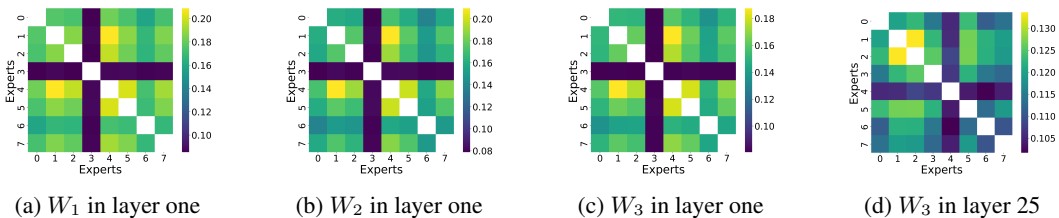

Figure 7: DOCS similarity scores between the MoE experts.

experiments uncovered meaningful patterns, including high similarity between neighboring layers and the presence of similar layer clusters. These findings underscore the potential of DOCS to guide applications in several ways. First, the identified cluster structures could be leveraged to introduce inductive biases during the supervised fine-tuning stage of parameter-efficient techniques. Second, the clustering results offer valuable information for designing sparsity patterns aimed at model compression. By targeting redundant connections within clustered layers, we may be able to significantly reduce model size and computational costs without compromising performance. Third, these findings can inform more efficient knowledge distillation strategies by identifying critical layers that deserve prioritization in the distillation process. A student model could focus on replicating the most representative layers within each cluster, thereby alleviating the computational overhead associated with emulating the teacher model's entire architecture.

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

## A PROOFS OF MATHEMATICAL PROPERTIES OF DOCS SIMILARITY INDICES

### A.1 PROOF OF PERMUTATION TRANSFORMATION INVARIANCE

**Lemma 1** (Permutation Transformation Invariance). *Let $X, Y \in \mathbb{R}^{n \times m}$ and let $P_X, P_Y \in \mathbb{R}^{m \times m}$ be permutation matrices. Then the DOCS similarity index is invariant under permutation transformations:*

$$S(X, Y) = S(XP_X, YP_Y).$$

*Proof.* We aim to show that the DOCS similarity index is invariant under permutation transformations:

$$S_{\text{DOCS}}(X, Y) = S_{\text{DOCS}}(XP_X, YP_Y),$$

where $P_X$ and $P_Y$ are permutation matrices of size $m \times m$.

First, recall that multiplying a matrix by a permutation matrix permutes its columns. Specifically, if $\sigma_X$ is the permutation associated with $P_X$, then:

$$XP_X = [X_{\sigma_X(1)}, X_{\sigma_X(2)}, \ldots, X_{\sigma_X(m)}],$$

where $X_{\sigma_X(j)}$ denotes the $\sigma_X(j)$-th column of $X$. Similarly, $YP_Y$ permutes the columns of $Y$ according to the permutation $\sigma_Y$:

$$YP_Y = [Y_{\sigma_Y(1)}, Y_{\sigma_Y(2)}, \ldots, Y_{\sigma_Y(m)}].$$

Next, compute the cosine similarity matrix $C \in \mathbb{R}^{m \times m}$ between $X$ and $Y$:

$$C_{jk} = \frac{X_j^\top Y_k}{\|X_j\| \, \|Y_k\|},$$

where $X_j$ and $Y_k$ are the $j$-th and $k$-th columns of $X$ and $Y$, respectively.

When we compute the cosine similarity matrix $C'$ between the permuted matrices $XP_X$ and $YP_Y$, we get:

$$C'_{jk} = \frac{(XP_X)_j^\top (YP_Y)_k}{\|(XP_X)_j\| \, \|(YP_Y)_k\|} = \frac{X_{\sigma_X(j)}^\top Y_{\sigma_Y(k)}}{\|X_{\sigma_X(j)}\| \, \|Y_{\sigma_Y(k)}\|}.$$

This shows that $C'_{jk} = C_{\sigma_X(j),\, \sigma_Y(k)}$. In other words, $C'$ is a reordering of the entries of $C$ based on the permutations $\sigma_X$ and $\sigma_Y$.

In the MAXCOSSIM function of the DOCS algorithm 1, for each $j$, we compute:

$$s_{X_j} = \max_k |C_{jk}|.$$

Similarly, for the permuted matrices, we have:

$$s_{(XP_X)_j} = \max_k |C'_{jk}| = \max_k |C_{\sigma_X(j),\, \sigma_Y(k)}|.$$

Since $\sigma_Y$ is a permutation of $\{1, 2, \ldots, m\}$, as $k$ ranges over 1 to $m$, so does $\sigma_Y(k)$. Therefore:

$$s_{(XP_X)_j} = \max_k |C_{\sigma_X(j),\, \sigma_Y(k)}| = \max_k |C_{\sigma_X(j),\, k}| = s_{X_{\sigma_X(j)}}.$$

This means that $s_{(XP_X)_j}$ is equal to $s_{X_{\sigma_X(j)}}$, indicating that the sequence $\{s_{(XP_X)_j}\}$ is a permutation of $\{s_{X_j}\}$ based on $\sigma_X$.

Similarly, for $Y$ and its permutation $YP_Y$, we find:

$$s_{(YP_Y)_k} = \max_j |C'_{kj}| = \max_j |C_{\sigma_X(j),\, \sigma_Y(k)}| = \max_j |C_{j,\, \sigma_Y(k)}| = s_{Y_{\sigma_Y(k)}}.$$

Thus, $s_{(YP_Y)_k}$ is equal to $s_{Y_{\sigma_Y(k)}}$, so the sequence $\{s_{(YP_Y)_k}\}$ is a permutation of $\{s_{Y_k}\}$ based on $\sigma_Y$.

When we fit Gumbel distributions to the sequences $\{s_{X_j}\}$ and $\{s_{(XP_X)_j}\}$, or $\{s_{Y_k}\}$ and $\{s_{(YP_Y)_k}\}$, the estimated location parameters $u_X$ and $u'_X$ (or $u_Y$ and $u'_Y$) remain the same because the sets of values are identical up to permutation.

Therefore, the DOCS similarity index remains unchanged:

$$S_{\text{DOCS}}(XP_X, YP_Y) = \frac{u'_X + u'_Y}{2} = \frac{u_X + u_Y}{2} = S_{\text{DOCS}}(X,Y).$$

This concludes the proof that the DOCS similarity index is invariant under permutation transformations. $\square$

### A.2 PROOF OF SYMMETRY

**Lemma 2** (Symmetry). *For any matrices $X, Y \in \mathbb{R}^{n \times m}$, the DOCS similarity index satisfies:*

$$S_{DOCS}(X,Y) = S_{DOCS}(Y,X).$$

*Proof.* Compute the cosine similarity matrix $C \in \mathbb{R}^{m \times m}$ where:

$$C_{jk} = \frac{X_j^\top Y_k}{\|X_j\| \, \|Y_k\|}.$$

Note that $C_{jk} = C_{kj}$ because:

$$C_{jk} = \frac{X_j^\top Y_k}{\|X_j\| \, \|Y_k\|} = \frac{Y_k^\top X_j}{\|Y_k\| \, \|X_j\|} = C_{kj}.$$

Therefore, the cosine similarity matrix $C$ is symmetric.

The MaxCosSim function computes for each $j$:

$$s_{X_j} = \max_k |C_{jk}|.$$

Similarly, for $Y$ and $X$:

$$s_{Y_k} = \max_j |C_{kj}| = \max_j |C_{jk}|.$$

Thus, the sets $\{s_{X_j}\}$ and $\{s_{Y_k}\}$ are identical.

Fitting Gumbel distributions to $\{s_{X_j}\}$ and $\{s_{Y_k}\}$ yields identical location parameters:

$$u_X = u_Y.$$

Therefore, the DOCS similarity index is:

$$S_{\text{DOCS}}(X,Y) = \frac{u_X + u_Y}{2} = u_X = u_Y = S_{\text{DOCS}}(Y,X).$$

$\square$

### A.3 PROOF OF ISOTROPIC SCALING INVARIANCE INVARIANCE

**Lemma 3** (Isotropic Scaling Invariance). *For any matrices $X, Y \in \mathbb{R}^{n \times m}$ and nonzero scalars $a, b \in \mathbb{R}$, the DOCS similarity index satisfies:*

$$S_{DOCS}(aX, bY) = S_{DOCS}(X,Y).$$

*Proof.* Consider the scaled matrices:

$$\tilde{X} = aX, \quad \tilde{Y} = bY.$$

The cosine similarity between columns $\tilde{X}_j$ and $\tilde{Y}_k$ is:

$$\tilde{C}_{jk} = \frac{\tilde{X}_j^\top \tilde{Y}_k}{\|\tilde{X}_j\| \, \|\tilde{Y}_k\|} = \frac{(aX_j)^\top (bY_k)}{\|aX_j\| \, \|bY_k\|} = \frac{ab X_j^\top Y_k}{a\|X_j\| \cdot b\|Y_k\|} = \frac{X_j^\top Y_k}{\|X_j\| \, \|Y_k\|} = C_{jk}.$$

Therefore, the cosine similarity matrix $\tilde{C}$ is identical to $C$.

For each column $\tilde{X}_j$, the maximum absolute cosine similarity is:

$$s_{\tilde{X}_j} = \max_k |\tilde{C}_{jk}| = \max_k |C_{jk}| = s_{X_j}.$$

Similarly, $s_{\tilde{Y}_k} = s_{Y_k}$.

Thus, the vectors $\mathbf{s}_{\tilde{X}}$ and $\mathbf{s}_X$ are the same, as are $\mathbf{s}_{\tilde{Y}}$ and $\mathbf{s}_Y$. Fitting a Gumbel distribution yields the same location parameters:

$$u_{\tilde{X}} = u_X, \quad u_{\tilde{Y}} = u_Y.$$

Therefore,

$$S_{\text{DOCS}}(aX, bY) = \frac{u_{\tilde{X}} + u_{\tilde{Y}}}{2} = \frac{u_X + u_Y}{2} = S_{\text{DOCS}}(X, Y).$$

$\square$

## A.4  PROOF OF REFLEXIVITY

**Lemma 4** (Reflexivity). *For any matrix $X \in \mathbb{R}^{n \times m}$, the DOCS similarity index satisfies:*

$$S_{DOCS}(X, X) = 1.$$

*Proof.* Compute the cosine similarity matrix $C \in \mathbb{R}^{m \times m}$ where:

$$C_{jk} = \frac{X_j^\top X_k}{\|X_j\| \, \|X_k\|}.$$

When $j = k$, we have:

$$C_{jj} = \frac{X_j^\top X_j}{\|X_j\|^2} = 1.$$

Since the absolute value of the cosine similarity is bounded by 1, for each $j$:

$$s_{X_j} = \max_k |C_{jk}| = 1.$$

Therefore, the vector $\mathbf{s}_X = [1, 1, \ldots, 1]^\top$.

Fitting a Gumbel distribution to $\mathbf{s}_X$ consisting of all ones yields a location parameter:

$$u_X = 1.$$

Similarly, since $Y = X$, we have $\mathbf{s}_Y = \mathbf{s}_X$ and $u_Y = u_X = 1$.

Therefore, the DOCS similarity index is:

$$S_{\text{DOCS}}(X, X) = \frac{u_X + u_Y}{2} = \frac{1 + 1}{2} = 1.$$

$\square$

## A.5  PROOF OF DISCRIMINATIVE ON ORTHOGONAL MATRICES

**Lemma 5** (Discriminative on Orthogonal Matrices). *There exist orthogonal matrices $X, Y, X', Y'$ such that:*

$$S_{DOCS}(X, Y) \neq S_{DOCS}(X', Y').$$

*Proof.* Consider the following orthogonal matrices:

First pair:

$$X = \begin{bmatrix} -0.6676 & 0.5171 & -0.5357 \\ -0.7310 & -0.5917 & 0.3399 \\ -0.1412 & 0.6185 & 0.7730 \end{bmatrix}, \quad Y = \begin{bmatrix} -0.1837 & 0.5950 & 0.7825 \\ 0.0457 & -0.7900 & 0.6114 \\ 0.9819 & 0.1481 & 0.1179 \end{bmatrix}.$$

Second pair:

$$X' = \begin{bmatrix} -0.8499 & 0.0164 & 0.5267 \\ -0.0816 & -0.9915 & -0.1009 \\ 0.5206 & -0.1287 & 0.8440 \end{bmatrix}, \quad Y' = \begin{bmatrix} -0.7028 & -0.6446 & -0.3009 \\ 0.3734 & -0.6943 & 0.6153 \\ -0.6056 & 0.3200 & 0.7286 \end{bmatrix}.$$

Compute the DOCS similarity index for each pair:

For $X$ and $Y$:

$$S_{\text{DOCS}}(X, Y) = 0.88.$$

For $X'$ and $Y'$:

$$S_{\text{DOCS}}(X', Y') = 0.76.$$

Since $S_{\text{DOCS}}(X, Y) \neq S_{\text{DOCS}}(X', Y')$, the DOCS similarity index distinguishes between different pairs of orthogonal matrices. Therefore, it is discriminative on orthogonal matrices. $\square$

## B    PROOFS OF NON-DISCRIMINATIVE NATURE OF OTHER SIMILARITY INDICES FOR ORTHOGONAL MATRICES

In this section, we present the proofs demonstrating that other common similarity indices are either constant or depend only on the dimensions of the matrices when applied to orthogonal matrices.

### B.1    LINEAR REGRESSION

The Linear Regression Similarity is defined as:

$$S_{\text{LR}}(X, Y) = \frac{\left\| Q_Y^{\text{T}} X \right\|_{\text{F}}^2}{\|X\|_{\text{F}}^2}$$

where $Q_X$ and $Q_Y$ are orthonormal bases for the columns of $X$ and $Y$, respectively.

**Lemma 6.** *For any orthogonal matrices $X, Y \in \mathbb{R}^{n \times n}$,*

$$S_{LR}(X, Y) = \frac{\left\| Q_Y^{\text{T}} X \right\|_{\text{F}}^2}{\|X\|_{\text{F}}^2}$$

*is a constant that is independent of $n$.*

*Proof.* Since $X$ and $Y$ are orthogonal matrices, their columns form orthonormal bases. Simply let $Q_X = X$ and $Q_Y = Y$.

We begin by evaluating the Frobenius norm squared of $Q_Y^{\text{T}} X$:

$$\left\| Q_Y^{\text{T}} X \right\|_{\text{F}}^2 = \left\| Y^{\text{T}} X \right\|_{\text{F}}^2 = \text{trace}\left((Y^{\text{T}} X)^{\text{T}} (Y^{\text{T}} X)\right) = \text{trace}\left(X^{\text{T}} Y Y^{\text{T}} X\right).$$

Since $Y$ is orthogonal, $YY^{\text{T}} = I$. Thus, the expression simplifies to:

$$\text{trace}\left(X^{\text{T}} X\right) = \text{trace}\left(I\right) = n.$$

Next, we compute the Frobenius norm squared of $X$:

$$\|X\|_{\text{F}}^2 = \text{trace}\left(X^{\text{T}} X\right) = \text{trace}\left(I\right) = n.$$

Therefore, the ratio is:

$$\frac{\left\| Q_Y^{\text{T}} X \right\|_{\text{F}}^2}{\|X\|_{\text{F}}^2} = \frac{n}{n} = 1.$$

$\square$

## B.2 Canonical Correlation Analysis (CCA) with $R^2_{\text{CCA}}$

CCA $\left(R^2_{\text{CCA}}\right)$ is defined as:

$$S_{\text{CCA}\left(R^2_{\text{CCA}}\right)}(X, Y) = \frac{\left\|Q_Y^{\mathrm{T}} Q_X\right\|_{\mathrm{F}}^2}{n},$$

where $Q_X$ and $Q_Y$ represent orthonormal bases corresponding to the columns of $X$ and $Y$, respectively.

**Lemma 7.** *For any orthogonal matrices $X, Y \in \mathbb{R}^{n \times n}$, the quantity*

$$S_{CCA\left(R^2_{\text{CCA}}\right)}(X, Y) = \frac{\left\|Q_Y^{\mathrm{T}} Q_X\right\|_{\mathrm{F}}^2}{n}$$

*is a constant that is independent of $n$.*

*Proof.* Since $X$ and $Y$ are orthogonal matrices, their columns form orthonormal bases. Therefore, $Q_X = X$ and $Q_Y = Y$.

Consider the matrix $Q_Y^{\mathrm{T}} Q_X = Y^{\mathrm{T}} X$. Since both $Y$ and $X$ are orthogonal, their product $Y^{\mathrm{T}} X$ is also an orthogonal matrix.

The Frobenius norm of an orthogonal matrix $Y^{\mathrm{T}} X$ is given by

$$\left\|Y^{\mathrm{T}} X\right\|_{\mathrm{F}}^2 = \text{trace}\left((Y^{\mathrm{T}} X)^{\mathrm{T}} (Y^{\mathrm{T}} X)\right) = \text{trace}(I_n) = n,$$

where $I_n$ is the $n \times n$ identity matrix.

Therefore,

$$\frac{\left\|Q_Y^{\mathrm{T}} Q_X\right\|_{\mathrm{F}}^2}{n} = \frac{n}{n} = 1.$$

$\square$

## B.3 Canonical Correlation Analysis (CCA) with $\bar{\rho}_{\text{CCA}}$

CCA $\left(\bar{\rho}_{\text{CCA}}\right)$ is defined as:

$$S_{CCA(\bar{\rho}_{\text{CCA}})}(X, Y) = \frac{\left\|Q_Y^{\mathrm{T}} Q_X\right\|_*}{n},$$

where $Q_X$ and $Q_Y$ represent orthonormal bases corresponding to the columns of $X$ and $Y$, respectively.

**Lemma 8.** *For any orthogonal matrices $X, Y \in \mathbb{R}^{n \times n}$,*

$$S_{CCA(\bar{\rho}_{\text{CCA}})}(X, Y) = \frac{\left\|Q_Y^{\mathrm{T}} Q_X\right\|_*}{n}$$

*is a constant that is independent of $n$.*

*Proof.* Since $X$ and $Y$ are orthogonal matrices, their columns form orthonormal bases. Therefore, $Q_X$ and $Q_Y$ are also orthogonal matrices.

The product $Q_Y^{\mathrm{T}} Q_X$ is an orthogonal matrix because the product of orthogonal matrices is orthogonal. For any orthogonal matrix $A$, the nuclear norm $\|A\|_*$ is equal to the sum of its singular values. Since all singular values of an orthogonal matrix are equal to 1, we have:

$$\left\|Q_Y^{\mathrm{T}} Q_X\right\|_* = \sum_{i=1}^n \sigma_i(Q_Y^{\mathrm{T}} Q_X) = \sum_{i=1}^n 1 = n$$

Therefore,

$$\frac{\left\|Q_Y^{\mathrm{T}} Q_X\right\|_*}{n} = \frac{n}{n} = 1.$$

$\square$

## B.4 SINGULAR VECTOR CCA (SVCCA) WITH $R^2_{\text{SVCCA}}$

SVCCA $\left(R^2_{\text{SVCCA}}\right)$ is defined as:

$$S_{R^2_{\text{SVCCA}}} = \frac{\left\| (U_Y T_Y)^{\text{T}} U_X T_X \right\|^2_{\text{F}}}{\min\left( \|T_X\|^2_{\text{F}}, \|T_Y\|^2_{\text{F}} \right)},$$

where $U_X$ and $U_Y$ represent the left singular vectors of $X$ and $Y$, respectively, arranged in descending order based on their associated singular values. Meanwhile, $T_X$ and $T_Y$ denote truncated identity matrices that retain the left singular vectors, ensuring that the accumulated variance meets a predefined limit. Here, we consider $T_X = T_Y = I$.

**Lemma 9.** *Assume that $T_X = T_Y = I$. For any orthogonal matrices $X, Y \in \mathbb{R}^{n \times n}$,*

$$S_{R^2_{\text{SVCCA}}} = \frac{\left\| (U_Y T_Y)^{\text{T}} U_X T_X \right\|^2_{\text{F}}}{\min\left( \|T_X\|^2_{\text{F}}, \|T_Y\|^2_{\text{F}} \right)}$$

*is a constant independent of $n$.*

*Proof.* Given that $T_X = T_Y = I$, the expression simplifies to

$$\frac{\left\| (U_Y I)^{\text{T}} U_X I \right\|^2_{\text{F}}}{\min\left( \|I\|^2_{\text{F}}, \|I\|^2_{\text{F}} \right)} = \frac{\left\| U_Y^{\text{T}} U_X \right\|^2_{\text{F}}}{\|I\|^2_{\text{F}}}.$$

Since $U_X$ and $U_Y$ are orthogonal matrices, their product $U_Y^{\text{T}} U_X$ is also an orthogonal matrix, denoted by $Q$. The Frobenius norm of an orthogonal matrix satisfies

$$\|Q\|^2_{\text{F}} = \sum_{i=1}^n \sum_{j=1}^n Q_{ij}^2 = \text{trace}(Q^{\text{T}} Q) = \text{trace}(I) = n.$$

Additionally, the Frobenius norm of the identity matrix $I$ is

$$\|I\|^2_{\text{F}} = \sum_{i=1}^n \sum_{j=1}^n I_{ij}^2 = \text{trace}(I) = n.$$

Substituting these results back into the original expression, we obtain

$$\frac{\left\| U_Y^{\text{T}} U_X \right\|^2_{\text{F}}}{\|I\|^2_{\text{F}}} = \frac{n}{n} = 1.$$

$\square$

## B.5 SINGULAR VECTOR CCA (SVCCA) WITH $\bar{\rho}_{\text{SVCCA}}$

SVCCA $(\bar{\rho}_{\text{SVCCA}})$ is defined as:

$$S_{\bar{\rho}_{\text{SVCCA}}} = \frac{\left\| (U_Y T_Y)^{\text{T}} U_X T_X \right\|_*}{\min\left( \|T_X\|^2_{\text{F}}, \|T_Y\|^2_{\text{F}} \right)},$$

where $U_X$ and $U_Y$ represent the left singular vectors of $X$ and $Y$, respectively, arranged in descending order based on their associated singular values. Meanwhile, $T_X$ and $T_Y$ denote truncated identity matrices that retain the left singular vectors, ensuring that the accumulated variance meets a predefined limit. Here, we consider $T_X = T_Y = I$.

**Lemma 10.** *Assume $T_X = T_Y = I$. For any orthogonal matrices $X, Y \in \mathbb{R}^{n \times n}$,*

$$S_{\bar{\rho}_{SVCCA}} = \frac{\left\| (U_Y T_Y)^{\mathrm{T}} U_X T_X \right\|_*}{\min \left( \|T_X\|_{\mathrm{F}}^2, \|T_Y\|_{\mathrm{F}}^2 \right)}$$

*is a constant independent of $n$.*

*Proof.* Since $X$ and $Y$ are orthogonal matrices in $\mathbb{R}^{n \times n}$, their singular value decompositions (SVDs) are given by

$$X = U_X \Sigma_X V_X^{\mathrm{T}}, \quad Y = U_Y \Sigma_Y V_Y^{\mathrm{T}},$$

where $\Sigma_X$ and $\Sigma_Y$ are diagonal matrices of singular values. Because $X$ and $Y$ are orthogonal, their singular values are all equal to 1; thus, $\Sigma_X = \Sigma_Y = I$. Therefore, the SVDs simplify to

$$X = U_X V_X^{\mathrm{T}}, \quad Y = U_Y V_Y^{\mathrm{T}}.$$

Given that $T_X = T_Y = I$, the expression reduces to

$$\frac{\left\| U_Y^{\mathrm{T}} U_X \right\|_*}{\min \left( \|I\|_{\mathrm{F}}^2, \|I\|_{\mathrm{F}}^2 \right)} = \frac{\left\| U_Y^{\mathrm{T}} U_X \right\|_*}{n}.$$

Since $U_X$ and $U_Y$ are orthonormal matrices, $U_Y^{\mathrm{T}} U_X$ is also an orthogonal matrix. The singular values of $U_Y^{\mathrm{T}} U_X$ are thus all equal to 1, and the nuclear norm is

$$\left\| U_Y^{\mathrm{T}} U_X \right\|_* = \sum_{i=1}^{n} \sigma_i = n.$$

Substituting back, we obtain

$$\frac{\left\| U_Y^{\mathrm{T}} U_X \right\|_*}{n} = \frac{n}{n} = 1.$$

Therefore, the quantity is a constant equal to 1, independent of $n$. $\qquad\square$

### B.6 LINEAR HILBERT-SCHMIDT INDEPENDENCE CRITERION (HSIC)

Linear HSIC is defined as:

$$S_{\text{Linear HSIC}}(X, Y) = \frac{\left\| Y^{\mathrm{T}} X \right\|_{\mathrm{F}}^2}{(n-1)^2}.$$

**Lemma 11.** *For any orthogonal matrices $X, Y \in \mathbb{R}^{n \times n}$,*

$$S_{\text{Linear HSIC}}(X, Y) = \frac{\left\| Y^{\mathrm{T}} X \right\|_{\mathrm{F}}^2}{(n-1)^2}$$

*is a constant that depends solely on $n$.*

*Proof.* Let $X$ and $Y$ be any orthogonal matrices in $\mathbb{R}^{n \times n}$. Since both $X$ and $Y$ are orthogonal, their transpose inverses satisfy $X^{\mathrm{T}} X = Y^{\mathrm{T}} Y = I_n$, where $I_n$ is the $n \times n$ identity matrix.

Consider the product $Y^{\mathrm{T}} X$. Since the product of two orthogonal matrices is also orthogonal, $Y^{\mathrm{T}} X$ is orthogonal. The Frobenius norm of an orthogonal matrix $A$ satisfies

$$\|A\|_{\mathrm{F}}^2 = \text{trace}(A^{\mathrm{T}} A).$$

Applying this to $Y^{\mathrm{T}} X$, we have

$$\left\| Y^{\mathrm{T}} X \right\|_{\mathrm{F}}^2 = \text{trace} \left( (Y^{\mathrm{T}} X)^{\mathrm{T}} Y^{\mathrm{T}} X \right) = \text{trace} \left( X^{\mathrm{T}} Y Y^{\mathrm{T}} X \right).$$

Since $YY^{\mathrm{T}} = I_n$, this simplifies to

$$\text{trace} \left( X^{\mathrm{T}} X \right) = \text{trace}(I_n) = n.$$

Therefore,

$$\frac{\left\| Y^{\mathrm{T}} X \right\|_{\mathrm{F}}^2}{(n-1)^2} = \frac{n}{(n-1)^2},$$

which depends only on $n$. This concludes the proof. $\qquad\square$

### B.7 LINEAR CENTERED KERNEL ALIGNMENT (CKA)

Linear CKA is defined as:

$$S_{\text{CKA}}(X, Y) = \frac{\|X^\top Y\|_F^2}{\|X^\top X\|_F \|Y^\top Y\|_F}.$$

**Lemma 12.** *For any orthogonal matrices $X, Y \in \mathbb{R}^{n \times n}$, the ratio*

$$\frac{\|Y^\text{T} X\|_\text{F}^2}{\|X^\text{T} X\|_\text{F} \|Y^\text{T} Y\|_\text{F}}$$

*is a constant independent of $n$.*

*Proof.* Since $X$ and $Y$ are orthogonal matrices, we have:

$$X^\text{T} X = I_n \quad \text{and} \quad Y^\text{T} Y = I_n,$$

where $I_n$ is the $n \times n$ identity matrix.

The Frobenius norm of the identity matrix is:

$$\|I_n\|_\text{F} = \sqrt{\sum_{i=1}^{n} \sum_{j=1}^{n} \delta_{ij}^2} = \sqrt{n},$$

where $\delta_{ij}$ is the Kronecker delta.

Thus, we have:

$$\|X^\text{T} X\|_\text{F} = \|I_n\|_\text{F} = \sqrt{n}, \quad \|Y^\text{T} Y\|_\text{F} = \|I_n\|_\text{F} = \sqrt{n}.$$

Next, consider the matrix $Y^\text{T} X$. Since both $X$ and $Y$ are orthogonal, $Y^\text{T} X$ is also orthogonal:

$$(Y^\text{T} X)^\text{T} (Y^\text{T} X) = X^\text{T} Y Y^\text{T} X = X^\text{T} X = I_n.$$

Therefore, the Frobenius norm of $Y^\text{T} X$ is:

$$\|Y^\text{T} X\|_\text{F} = \|I_n\|_\text{F} = \sqrt{n}.$$

Substituting these results into the original ratio, we obtain:

$$\frac{\|Y^\text{T} X\|_\text{F}^2}{\|X^\text{T} X\|_\text{F} \|Y^\text{T} Y\|_\text{F}} = \frac{(\sqrt{n})^2}{\sqrt{n} \cdot \sqrt{n}} = \frac{n}{n} = 1.$$

Therefore, the ratio is equal to 1, which is a constant independent of $n$.

$\square$

## C PROOF OF THEOREM 1

We construct specific matrices $X$ and $Y$ to verify the stated properties.

**Construction of $X$ and $Y$:**

Let $n \geq 2$. Let $m = 2^{\lfloor \log_2 n \rfloor}$, so $m = \Omega(n)$. Define:

- $X \in \mathbb{R}^{n \times m}$ as $X = [e_1, e_2, \ldots, e_m]$, where $e_i$ is the $i$-th standard basis vector in $\mathbb{R}^n$.
- $Y \in \mathbb{R}^{n \times m}$ as:
$$Y = \frac{1}{\sqrt{m}} \begin{pmatrix} H_m \\ \mathbf{0}_{(n-m) \times m} \end{pmatrix},$$
  where $H_m$ is the $m \times m$ Hadamard matrix, and $\mathbf{0}$ is a zero matrix.

**Orthogonality of $X$ and $Y$:**

- The columns of $X$ are orthonormal since they are standard basis vectors.
- For $Y$:

$$Y^\top Y = \left(\frac{1}{\sqrt{m}}H_m\right)^\top \left(\frac{1}{\sqrt{m}}H_m\right) = \frac{1}{m}H_m^\top H_m = I_m,$$

since $H_m H_m^\top = mI_m$. Thus, the columns of $Y$ are orthonormal.

**Computing $\|X - Y\|_F$:**

Compute the Frobenius norm:

$$\|X - Y\|_F^2 = \sum_{i=1}^{n}\sum_{j=1}^{m}\left(X_{ij} - Y_{ij}\right)^2.$$

For $1 \le i \le m$:

$$X_{ij} - Y_{ij} = \delta_{ij} - \frac{h_{ij}}{\sqrt{m}},$$

where $\delta_{ij}$ is the Kronecker delta and $h_{ij} = \pm 1$.

For $m + 1 \le i \le n$:

$$X_{ij} - Y_{ij} = 0 - 0 = 0.$$

**Diagonal terms** ($i = j$):

$$\left(1 - \frac{h_{ii}}{\sqrt{m}}\right)^2 = \left(1 \mp \frac{1}{\sqrt{m}}\right)^2 = 1 \mp \frac{2}{\sqrt{m}} + \frac{1}{m}.$$

**Off-diagonal terms** ($i \ne j, 1 \le i, j \le m$):

$$\left(0 - \frac{h_{ij}}{\sqrt{m}}\right)^2 = \frac{1}{m}.$$

**Summing all terms:**

Number of diagonal terms: $m$. Number of off-diagonal terms: $m(m - 1)$.

**Diagonal sum**:

Since the $\mp \dfrac{2}{\sqrt{m}}$ terms cancel out when summed over all $i$:

$$S_{\text{diag}} = m\left(1 + \frac{1}{m}\right) = m + 1.$$

**Off-diagonal sum**:

$$S_{\text{off-diag}} = m(m - 1) \times \frac{1}{m} = m - 1.$$

**Total Frobenius norm:**

$$\|X - Y\|_F^2 = S_{\text{diag}} + S_{\text{off-diag}} = (m + 1) + (m - 1) = 2m.$$

$$\|X - Y\|_F = \sqrt{2m} = \Omega(\sqrt{n}).$$

**Computing $S_{\text{DOCS}}(X, Y)$:**

For columns $X_j$ and $Y_k$, the cosine similarity is:

$$\cos(\theta_{jk}) = \frac{X_j^\top Y_k}{\|X_j\|\|Y_k\|} = \frac{Y_{jk}}{\|Y_k\|}.$$

Since $\|X_j\| = 1$ and $\|Y_k\| = 1$, and $Y_{jk} = \dfrac{h_{jk}}{\sqrt{m}}$, we have:

$$|\cos(\theta_{jk})| = \frac{1}{\sqrt{m}}.$$

Thus, the maximum absolute cosine similarity for each column is $\dfrac{1}{\sqrt{m}}$, so:

$$S_{\text{DOCS}}(X, Y) = \frac{1}{\sqrt{m}}.$$

This demonstrates that the DOCS similarity is inversely proportional to the square root of the number of columns, $m$.

## D  EXCLUDED MATHEMATICAL PROPERTIES

In this section, we discuss the additional mathematical properties that could be considered when evaluating similarity indices:

- **Invertible Linear Transformation (ILT) Invariance (Raghu et al., 2017)**:
$$S(X, Y) = S(XA, YB)$$
where $A$ and $B$ are arbitrary invertible matrices. This property describes the ability of an index to remain invariant under invertible linear transformations. It is not applicable for evaluating weight similarity measures, as weight matrices are not expected to undergo arbitrary invertible transformations. Similarly, it does not apply to representation similarity measures (Kornblith et al., 2019).

- **Translation (TR) Invariance (Raghu et al., 2017; Klabunde et al., 2023a)**:
$$S(X, Y) = S\left(X + \mathbf{1}\mathbf{c}^\top, Y + \mathbf{1}\mathbf{d}^\top\right)$$
where $\mathbf{c}$ and $\mathbf{d}$ are arbitrary constant vectors. This property describes the ability of an index to be unaffected by additive shifts in the data. It is not applicable for evaluating weight similarity measures because translating the parameters would result in a fundamentally different set of parameters, which is inconsistent for evaluation purposes.

- **Affine Transformations (AT) Invariance**:
$$S(X, Y) = S\left(XA + \mathbf{1}c^\top, YB + \mathbf{1}d^\top\right)$$
where $A$ and $B$ are arbitrary invertible matrices and $\mathbf{c}$, $\mathbf{d}$ are arbitrary constant vectors. This property is a combination of linear transformations and translations and is not applicable for evaluating weight similarity measures for the same reasons as ILT and TR Invariance.

# E    FURTHER DISCUSSIONS ON THE MOTIVATIONS OF DOCS

In this section, we provide additional justifications regarding (i) why we focus on *weight similarity* rather than *representational similarity*, (ii) the orthogonality of weight matrices in large language models, and (iii) the advantages of DOCS compared to other existing similarity indices.

## E.1    WEIGHT SIMILARITY VS. REPRESENTATIONAL SIMILARITY

As introduced in Section 1, while prior research has explored many methods for characterizing the similarity of neural networks (Wu et al., 2020; Khosla & Williams, 2024; Kriegeskorte et al., 2008; Klabunde et al., 2023b; Wang et al., 2020; Barannikov et al., 2021; Hamilton et al., 2016; Rahamim & Belinkov, 2024; Tang et al., 2020; Camastra & Staiano, 2016; Wang et al., 2018; Raghu et al., 2017; Morcos et al., 2018; Kornblith et al., 2019), these methods often focus on *representational similarity* rather than *weight similarity*.

Similar representations across layers do not necessarily imply similar weight matrices. This discrepancy arises from the use of *residual connections* in transformer architectures (He et al., 2016). This is evidenced by Figures 1a and 1b, which show that the input and output of the feedforward network have similar patterns of representational similarity

Here, we present another example where weight similarity reveals an intriguing finding that would otherwise be overlooked. We plot both the representational and weight similarity indices for the 01-ai/Yi-1.5-9B-Chat model. On one hand, we use the Linear CKA method to calculate the similarity between the outputs of MLP-UP layers. On the other hand, we employ the DOCS method to calculate the weight similarity of the MLP-UP layers. The experimental results are shown in Figure 8.

The results underscore the strengths of DOCS in revealing the weight structure. Specifically, DOCS reveals an intricate pattern in Figure 8a, indicating that layer 9 is highly similar to layer 25 , layer 10 is highly similar to layer 26, layer 11 is highly similar to layer 27, and so on. This suggests a repetition of a section of layers within 01-ai/Yi-1.5-9B-Chat, possibly due to a specific training strategy employed to save training costs.

In contrast, Figures 8b, 8c, and 8d present the representational similarity computed by Linear CKA using different input sequences. Despite using multiple input sentences for the analysis, the results provide a limited understanding of the weight structure. They exhibit relatively homogeneous patterns, lacking the fine-grained layer correspondence seen with DOCS. Therefore, in this example, focusing on weight similarity deepens our understanding of the model.

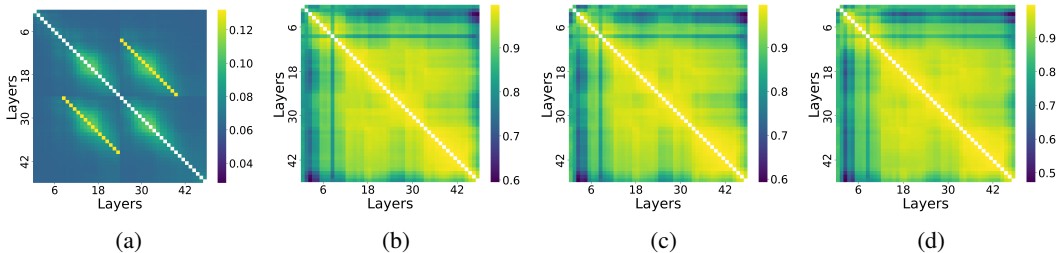

|        |        |        |        |
|--------|--------|--------|--------|
| (a)    | (b)    | (c)    | (d)    |

Figure 8: Comparison between DOCS and Linear CKA on the MLP-UP layers of the 01-ai/Yi-1.5-9B-Chat model. (a) shows the DOCS weight similarity of MLP-UP parameters across layers, while (b), (c), and (d) illustrate the Linear CKA representational similarity of the corresponding layer outputs with different input sentences.

## E.2    FURTHER EXPERIMENTS ON THE NON-DISCRIMINATIVE NATURE OF OTHER SIMILARITY INDICES

In Section 1, we emphasized that many existing similarity indices, such as Canonical Correlation Analysis (CCA) (Ramsay et al., 1984; Morcos et al., 2018), Singular Vector Canonical Correlation Analysis (SVCCA) (Raghu et al., 2017), and Linear Centered Kernel Alignment (Linear CKA)

(Kornblith et al., 2019), are *non-discriminative for orthogonal matrices*. An *orthogonal matrix* $Q$ is defined by the property $Q^\top Q = QQ^\top = I$, where $I$ is the identity matrix. This non-discriminative nature means that these indices can yield the same score when assessing the similarity between any two orthogonal matrices, regardless of their actual differences. These conclusions are presented in Section 2, with proofs provided in Appendix B. This issue is particularly relevant in the context of LLMs, where orthogonal matrices commonly occur throughout the training process (Tian et al., 2023). In fact, in Figure 2, we observe that when CKA is directly used to measure weight similarity, a large number of values concentrate in the high range of 0.78–0.80. This is likely due to the non-discriminative property for orthogonal matrices.

Here, we present additional results demonstrating a substantial degree of orthogonality among the weight matrices in LLMs. we introduce an index named the *Off-Diagonal Average Cosine Similarity* metric. This metric provides a systematic way to measure the extent of orthogonality between the columns of a given weight matrix. The metric is defined as:

$$
\text{Off-Diagonal Average Cosine Similarity}(\mathbf{X}) = \frac{1}{n(n-1)} \sum_{i=1}^{n} \sum_{\substack{j=1 \\ j \neq i}}^{n} \left| \frac{\mathbf{x}_i \cdot \mathbf{x}_j}{\|\mathbf{x}_i\| \, \|\mathbf{x}_j\|} \right|,
$$

where $\mathbf{X}$ is a matrix with $n$ columns, and $\mathbf{x}_i$ represents the $i$-th column of $\mathbf{X}$. This formulation indicates that a larger Off-Diagonal Average Cosine Similarity value corresponds to a lower degree of orthogonality in the matrix.

To further understand the orthogonality of weight matrices in LLMs, we constructed a family of approximately orthogonal matrices, denoted as $\mathcal{M}_\theta$, defined by:

$$
\mathcal{M}_\theta = I_n + \theta \, v \, \mathbf{1}^\top,
$$

where:

- $I_n$ is the identity matrix of size $n \times n$.

- $\theta \in \mathbb{R}$ is a scalar controlling the deviation from orthogonality; smaller values of $\theta$ correspond to matrices closer to orthogonal.

- $v \in \mathbb{R}^n$ is a random vector sampled from the normal distribution $v \sim \mathcal{N}(0, 1)$.

- $\mathbf{1} \in \mathbb{R}^n$ is a vector of ones.

In our study, we utilized the Q matrices and O matrices from each layer of the Meta-Llama-3.1-8B-Instruct model. Additionally, we generated four sets of approximately orthogonal matrices by sampling $\mathcal{M}_\theta$ with $\theta$ values of 0.005, 0.003, 0.002, and 0.001, ensuring that their shapes matched those of the Q and O matrices. We then computed the Off-diagonal Average Cosine Similarity for these matrices.

The experimental results, presented in Figures 9, reveal that the majority of the Q and O matrices from each layer of the Meta-Llama-3.1-8B-Instruct model exhibit stronger orthogonality compared to $\mathcal{M}_{0.003}$. According to the definition of $\mathcal{M}_\theta$, $\mathcal{M}_{0.003}$ only has a very small perturbation added to the identity matrix. This observation suggests that the Q and O matrices in the Meta-Llama-3.1-8B-Instruct model are highly orthogonal.

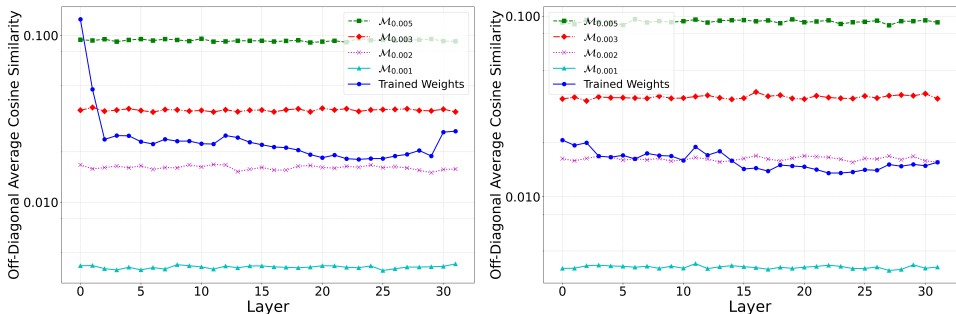

Figure 9: Average cosine similarity for Meta-Llama-3.1-8B-Instruct, with the Q matrix on the left and the O matrix on the right.

### E.3 ADVANTAGES OF DOCS OVER OTHER SIMILARITY INDICES

In Section 4.1, Figure 2 showcases the evaluation outcomes of eight different similarity indices on the MLP-UP layers from the Meta-Llama-3.1-8B-Instruct model. The resulting heatmaps highlight that indices like Linear Regression, Canonical Correlation Analysis (CCA), and CCA (Nuclear) fail to display discernible structural patterns. On the other hand, methods such as Singular Vector CCA (SVCCA), SVCCA (Nuclear), and Linear Centered Kernel Alignment (Linear CKA) exhibit faint block-like or striped formations in the off-diagonal regions, which might be attributed to either noise or inherent limitations in the indices themselves. These anomalies could arise due to the reduced sensitivity of these indices in differentiating between orthogonal matrices, as elaborated in Section 2.

To further compare the effectiveness of the DOCS method with other approaches, we conducted additional experiments using three models:

(A) `meta-llama/Meta-Llama-3.1-8B` (the base model)

(B) `meta-llama/Meta-Llama-3.1-8B-Instruct` (an instruction-tuned version of the base model)

(C) A version of `meta-llama/Meta-Llama-3.1-8B` with randomly initialized weights

Intuitively, the weight matrices of corresponding layers in models (A) and (B) should exhibit significantly higher similarity compared to those between models (A) and (C), since model (C) contains random weights and lacks the learned structure present in models (A) and (B).

To quantify this, we define the *similarity ratio* as the ratio of the similarity scores between models (A) and (B) to those between models (A) and (C). Specifically, for each similarity index, we compute the similarity of the corresponding MLP-UP and MLP-DOWN weight matrices between models (A) and (B), and between models (A) and (C). A higher ratio indicates that the similarity index is better at distinguishing between meaningful relationships in model weights (as seen in models (A) and (B)) and unrelated weights (as seen in models (A) and (C)). Thus, a higher ratio reflects the ability of the index to highlight structural patterns specific to related models while minimizing noise from uncorrelated data.

As shown in Figure 10, the experimental results reveal that the similarity ratio computed using the DOCS method is much higher—approximately 10 times greater—than those obtained with the other indices. This demonstrates the effectiveness of DOCS in capturing meaningful similarity patterns in model weights.

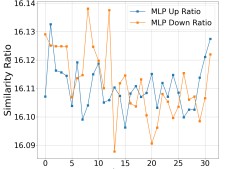 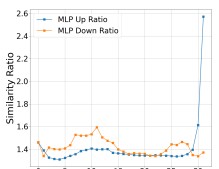 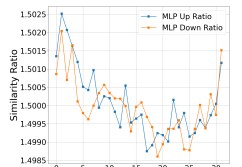 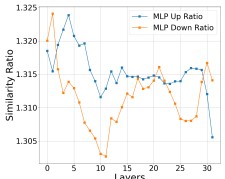

(a) Similarity ratio computed using the DOCS method.  (b) Similarity ratio computed using the Linear CKA method.  (c) Similarity ratio computed using the SVCCA method.  (d) Similarity ratio computed using the SVCCA (nuclear) method.

Figure 10: Comparison of similarity ratios between models (A) and (B) relative to models (A) and (C) across four methods: DOCS, Linear CKA, SVCCA, and SVCCA (nuclear). The DOCS method demonstrates a significantly higher ratio, indicating more effective performance.

## F  COMPARISON OF SIMILARITY INDICES ON A RANDOMLY INITIALIZED MODEL

Figure 11 provides a visual comparison of eight different similarity indices applied to the MLP-UP layers of the reinitialized Meta-Llama-3.1-8B-Instruct model. We observed that all of them exhibit relatively random patterns.

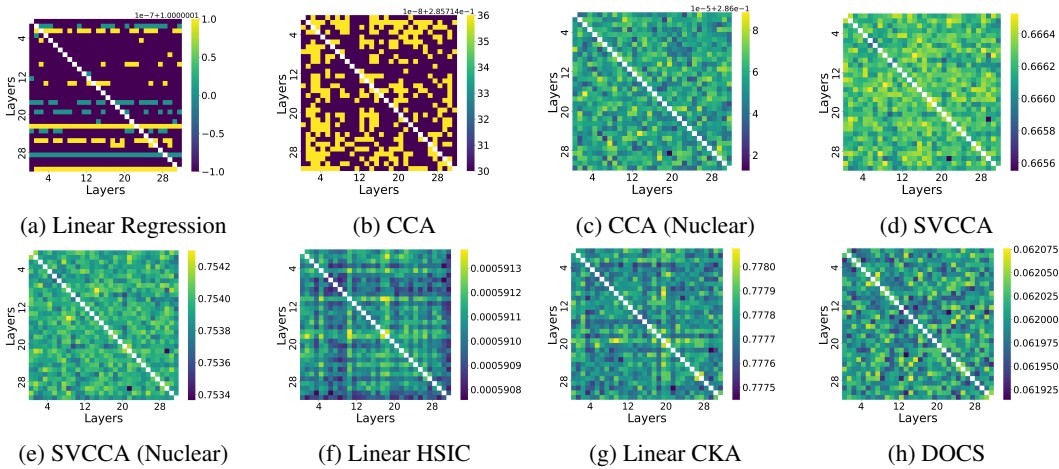

Figure 11: Comparison of similarity indices on the MLP-UP layers of reinitialized Meta-Llama-3.1-8B-Instruct.

## G  JUSTIFICATIONS FOR SPECIFIC STEPS IN THE DOCS ALGORITHM

In this section, we provide justifications for specific design choices made in the DOCS algorithm, focusing on the use of the Gumbel distribution and the maximization function in computing the similarity index.

**Justification for the Choice of Gumbel Distribution**

To illustrate why the Gumbel distribution is suitable for modeling the data in the DOCS algorithm, we analyze the distribution of maximum cosine similarities between columns of weight matrices from different layers.

Let $X$ be the MLP-UP weight matrix from the 4th layer, and let $Y$ be the MLP-UP weight matrix from the 8th layer of the `meta-llama/Meta-Llama-3.1-8B` model. We compute the vectors $\mathbf{s}_X = \text{MAXCOSSIM}(X, Y)$ and $\mathbf{s}_Y = \text{MAXCOSSIM}(Y, X)$, where MAXCOSSIM refers to the

function that computes, for each column in one matrix, the maximum absolute cosine similarity with any column in the other matrix.

The histograms of $\mathbf{s}_X$ and $\mathbf{s}_Y$ are plotted in Figure 12. As observed, the distributions of the maximum cosine similarities closely resemble the Gumbel distribution, which is commonly used to model the distribution of extreme values (maxima or minima) of samples of random variables. This empirical observation supports our choice of fitting a Gumbel distribution to the maximum cosine similarities in the DOCS algorithm.

By fitting a Gumbel distribution to these maximum similarity values and using the location parameter $u$ as the similarity index, we capture the central tendency of the extreme similarities in a way that is robust to outliers and sensitive to the most significant alignments between the weight matrices.

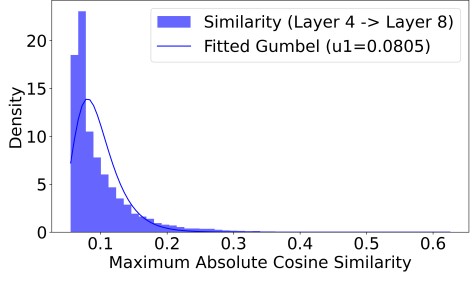
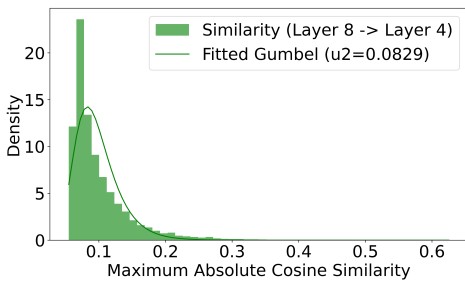

(a) Histogram of $\mathbf{s}_X$ (Layer 4 to Layer 8).      (b) Histogram of $\mathbf{s}_Y$ (Layer 8 to Layer 4).

Figure 12: Histograms of the maximum cosine similarity vectors $\mathbf{s}_X$ and $\mathbf{s}_Y$ for the MLP-UP weight matrices from layers 4 and 8. The overlaid curves represent the fitted Gumbel distributions.

**Importance of the Maximization Function**

The maximization operation in the MAXCOSSIM function plays a crucial role in the DOCS algorithm. To demonstrate this, we compare the results of the standard DOCS algorithm with a variant where the maximum operation is replaced by averaging.

In the standard DOCS algorithm, for each column in matrix $X$, we compute the maximum absolute cosine similarity with any column in matrix $Y$. This approach emphasizes the strongest alignments between the weight matrices, which are essential for detecting structural similarities.

Alternatively, using the average of the absolute cosine similarities incorporates all pairwise similarities, including weaker alignments. This averaging process can dilute the impact of the most significant similarities by combining them with less relevant ones.

Figure 13 compares the similarity heatmaps generated by the average-based DOCS and the standard DOCS algorithm for the MLP-UP parameter matrices of the `meta-llama/Meta-Llama-3.1-8B` model.

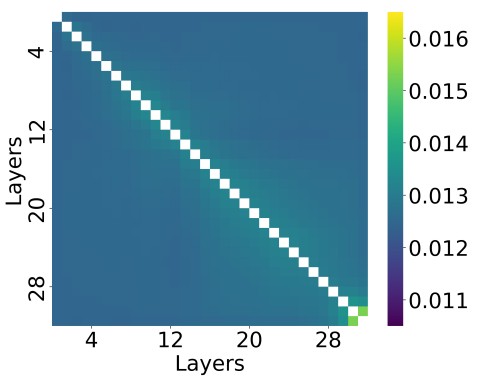

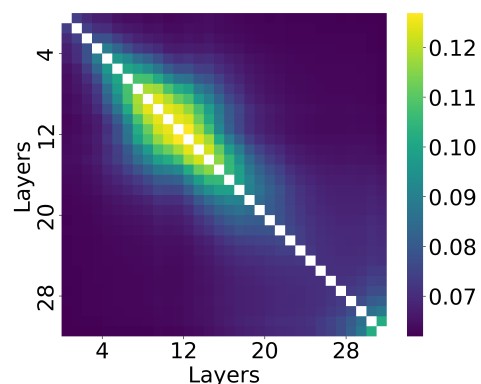

(a) Similarity heatmap using average-based DOCS.

(b) Similarity heatmap using standard DOCS algorithm.

Figure 13: Comparison of similarity heatmaps for the MLP-UP parameter matrices of the `meta-llama/Meta-Llama-3.1-8B` model: (a) average-based DOCS and (b) standard DOCS algorithm.

As shown in Figure 13, the similarity heatmap generated by the average-based DOCS lacks clear structural patterns and appears more uniform. In contrast, the heatmap produced by the standard DOCS algorithm reveals distinct structural similarities between specific layers, evident through pronounced diagonal and off-diagonal patterns.

This comparison highlights the importance of using the maximum operation in the MAXCOSSIM function. By focusing on the strongest relationships between columns of the weight matrices, the DOCS algorithm effectively captures meaningful structural patterns that might be obscured when using an averaging approach. The maximum operation ensures that significant alignments are emphasized, enhancing the sensitivity of the similarity measure to relevant features in the model's architecture.

## H COMPUTATION DETAILS OF GINI COEFFICIENT

The computation of the Gini coefficient involves several steps. First, we exclude the diagonal elements from the similarity matrix as they are always equal to one, focusing solely on inter-layer relationships. For each row, the remaining elements are normalized by dividing them by the sum of all row values, ensuring that the resulting coefficients reflect the relative distribution of similarity scores within the row. The Gini coefficient for each row is then calculated to quantify the inequality in the distribution of these scores, with higher values indicating a greater concentration of similarity among fewer layer pairs. Finally, the mean Gini coefficient across all rows is computed to summarize the overall inequality in the similarity matrix.

A higher Gini coefficient suggests a more uneven distribution of similarity scores, implying a greater concentration of significant similarities in fewer layer pairs, which potentially reveals structural characteristics of the model parameters. As demonstrated in Table 2, our proposed method (DOCS) achieves higher Gini coefficients compared to other methods, indicating its ability to isolate significant layer similarities.

## I HEATMAPS OF DOCS SCORES ON VARIOUS LLMS

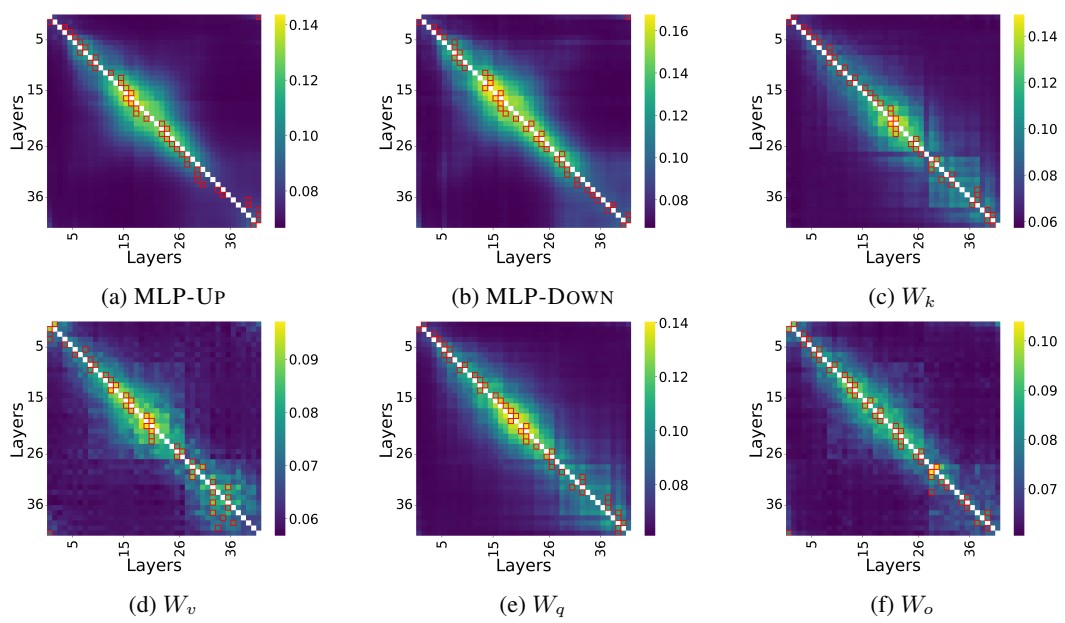

Figure 14: DOCS scores between transformer layers in gemma-2-9b.

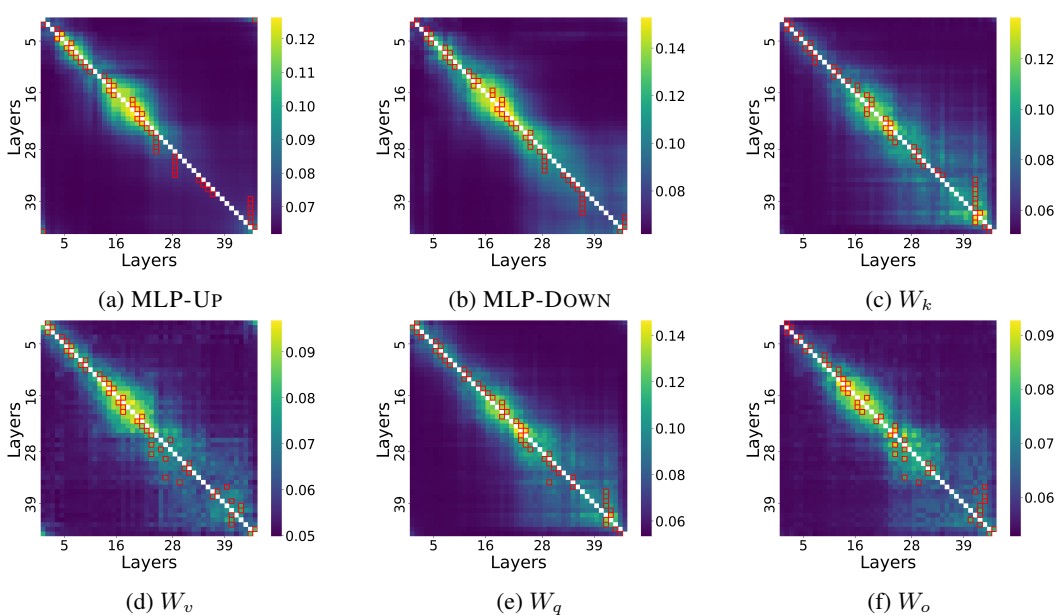

Figure 15: DOCS scores between transformer layers in gemma-2-27b.

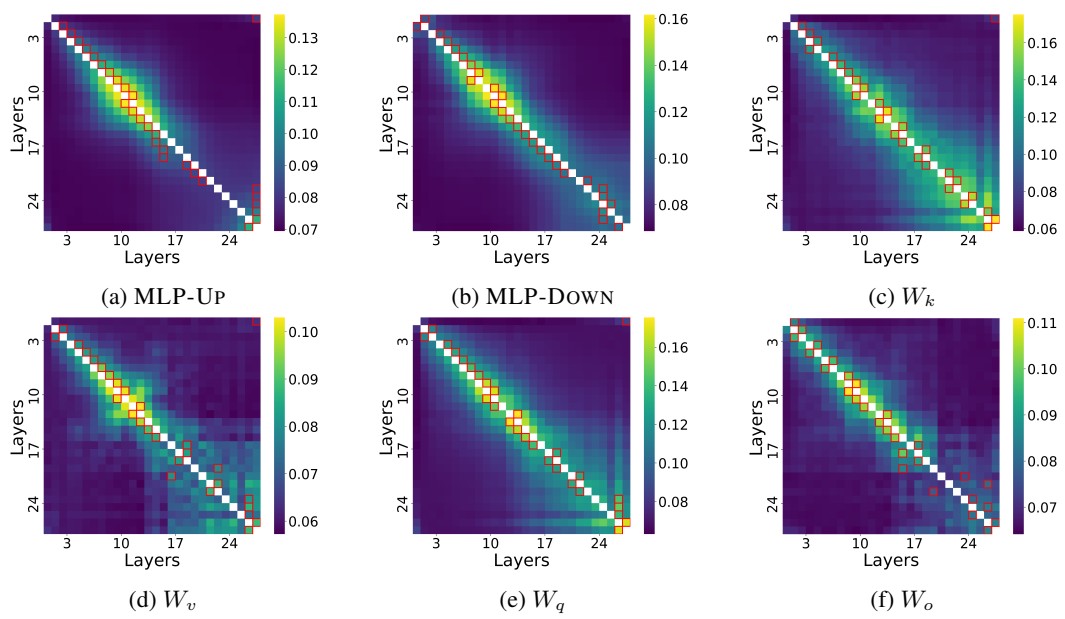

Figure 16: DOCS scores between transformer layers in Llama-3.1-3B.

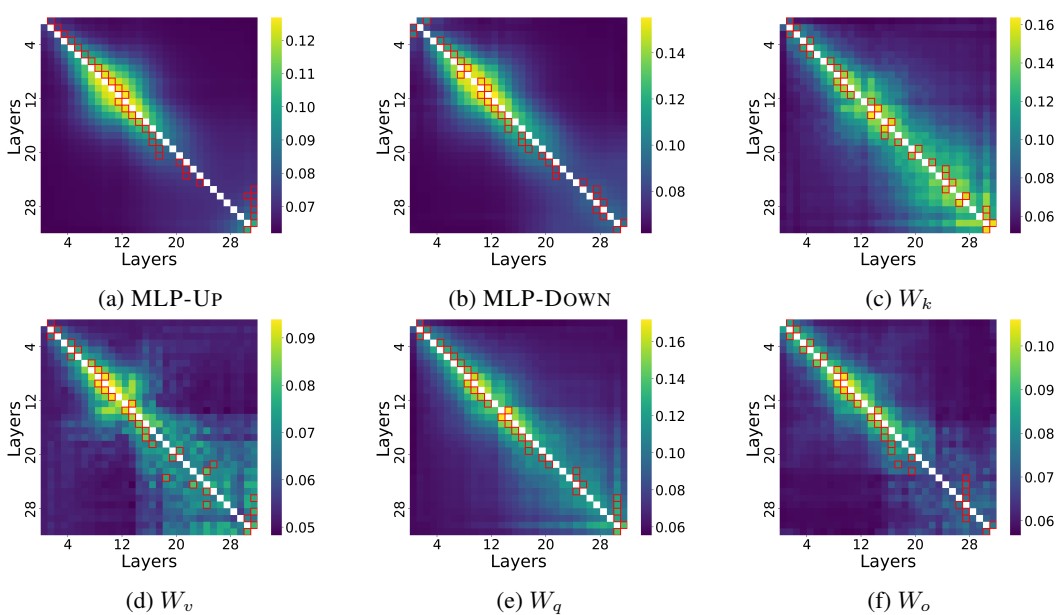

Figure 17: DOCS scores between transformer layers in Llama-3.1-8B.

