# OpenReview forum: "DOCS: Quantifying Weight Similarity for Deeper Insights into Large Language Models"
_ICLR.cc/2025/Conference — ICLR 2025 Poster_

### Official Review · Reviewer_37uY · 2024-10-16

**Soundness:** 3
**Presentation:** 3
**Contribution:** 3
**Rating:** 8
**Confidence:** 4

**Summary:**

This paper proposes a novel similarity index, DOCS, designed to measure the similarity between two matrices, with a focus on its application to neural network matrices. A key feature of DOCS is its ability to distinguish between orthogonal matrices, which the authors claim is crucial for training neural networks. In addition, DOCS possesses desirable properties such as symmetry and reflexivity.

The computation of DOCS is both simple and elegant, making it easy to understand and computationally efficient. It calculates the maximum cosine similarity between the rows and columns of two matrices, then fits a Gumbel distribution to each of these values. Finally, it averages the two scores to produce the final similarity measure.

The authors present a series of experiments demonstrating the results of applying DOCS to several large language models, comparing the results to other similarity indices such as CCA and SVCCA.

Overall, the paper introduces a promising new similarity index, DOCS, with several interesting properties for comparing neural network weight matrices, particularly in the context of language models. The authors provide an extensive set of experiments with recent open-weight models, comparing DOCS with other indices. However, while the contributions are valid, there are several issues that need to be addressed in the current version of the paper:

Evaluation
------

It is unclear how to evaluate DOCS or whether it captures a truly desirable property. The experiments in the paper show results, but it is not clear why DOCS should be considered superior to other similarity indices. How should we interpret these results? Are the similarities computed by DOCS more meaningful than those computed by other indices? The paper does not provide sufficient justification for why the "elegant" similarity matrices obtained by DOCS are more informative or useful than those obtained by other methods. The authors also suggest that DOCS could help in developing more efficient models (L.124) or provide deeper insights into model behavior (L.134), but there is no empirical evidence to support these claims.
Providing some empirical evidence that using DOCS can lead to better empirical results, or new insights about models are crucial as part of the evaluation of this method.

The authors attempt to justify DOCS by presenting heatmaps of weight matrices, suggesting that these show clearer patterns with less noise. However, this is a highly subjective measure and not convincing. They also compute the Gini coefficient (which should be explained more thoroughly) and claim that its higher value supports DOCS’s superiority. I find this reasoning unconvincing.


Overclaiming
--------
Several claims made in the paper are either overstated or lack sufficient evidence.

- The assertion that neural network matrices require a similarity index that discriminates between orthogonal matrices is one such claim. The authors should (1) empirically demonstrate that the matrices are orthogonal, which should be straightforward, and (2) show that this property is important in practice. Currently, these claims are made without adequate justification.
- L93-98: The authors claim that the heatmaps demonstrate the effectiveness of DOCS. Why do these heatmaps prove effectiveness? Why should we expect the DOCS index to be more informative than other similarity indices?
- L.113: The paper claims that DOCS shows high similarity, but the values are around 0.12, which seems quite low to me.
- L.124: The statement that DOCS "could lead to more efficient models" feels like an overreach.
- L.420-421: The claim that large language models (LLMs) might consist of multiple reasoning phases based on the similarity clusters is speculative and not well-substantiated.
- L.424: The argument that multiple clusters imply multiple training strategies is unconvincing. This could easily be tested by comparing models at different training stages (e.g., base model, fine-tuned model, RLHF model) and examining whether the number of clusters changes.
- L.477: The trends in similarity scores across different layers are minimal (e.g., differences of <0.001 in Llama-3.1-8B), so I question how meaningful these differences really are.


Missing Details
---------
(Note: these are smaller issues, but still important to address in the paper.)

- When computing DOCS, part 3 of the algorithm involves fitting a Gumbel distribution to the obtained values. However, it is unclear how the final single numeric value is derived. Is it the mean, median, or another statistic from the distribution?
- The claim that DOCS can distinguish between orthogonal matrices (L.267) is not explicitly demonstrated in the paper. More details should be provided.
- There is no information on how the baselines for comparison were computed. These details should be included for transparency.


Need for Elaboration
-------------
The paper discusses at length DOCS’s ability to discriminate between orthogonal matrices, but it is unclear why this is a valuable property. Expanding on why this distinction matters in practice would strengthen the paper.

The theoretical proof in Section 3.1 is also not entirely clear. It would be helpful if the authors could expand on the practical implications of this proof.


-------
Rebuttal summary:

After the authors' response I decided to increase my scores due to the satisfying responses that satisfied most of my concerns.
I still believe it's not completely clear how the metric can be used, but I believe some may find it useful, and I do not object of having another similarity metric, especially since it operates on matrices, unlike most previous work that operates on hidden representations.

**Strengths:**

- Introduction of a new similarity index, DOCS, which possesses intriguing properties that make it well-suited for comparing neural network similarities.
- Extensive experimentation on recent language models, with comparisons to established similarity indices.

**Weaknesses:**

- The evaluation of DOCS is insufficient and unconvincing, as discussed in detail in the review.
- Several claims made in the paper are overstated or lack adequate justification (see the summary section for further elaboration).
- Certain important details are missing or insufficiently explained (further discussed in the summary section).

**Questions:**

- It could be valuable to experiment with an initialized model and compute similarity indices, including DOCS. Given the presence of residual connections, the results may exhibit non-random patterns.
- L.115: This sentence is unclear. Are you suggesting that the layer configurations remain uniform during the initialization of the model?
- Suggestion: The font sizes in Figure 3 panels (1-3) and (d-f) are inconsistent. Ensure uniform font sizes across the figures for clarity and visual consistency.

---

> ### Author Response · Authors · 2024-11-23
>
> ## Evaluation and Problems 1 & 2 in Overclaiming
> We appreciate the reviewer's for raising these concerns. We have already addressed these in the general reply and added the content to Appendix E in the updated manuscript.
>
> ## Overclaiming Problem 3
> We apologize for the confusion caused by our original phrasing. The statement "exhibiting high mutual similarity" has been revised to "exhibiting relatively high mutual similarity (approximately twice the DOCS index values of other elements)" in the updated version of the manuscript.
>
> ## Overclaiming Problem 4
> We acknowledge the potential for confusion in our wording. The phrase "could lead to more efficient models" has been changed to "DOCS similarity can guide such designs" to clarify our intended meaning.
>
> ## Overclaiming Problem 5
> We appreciate your insightful comments and have revised the manuscript accordingly. Specifically, we have removed the discussion regarding the relationship between clusters within the LLM layers and reasoning phases, recognizing that it was speculative.
>
> ## Overclaiming Problem 6
> We appreciate the reviewer’s suggestion. Here, our intent was to convey that the distribution of clusters differs between models (e.g., Gemma-2-9b and Llama-3.1-70B), likely due to differences in their training strategies, rather than implying that a single model undergoes multiple training phases. This clarification has been incorporated into the revised manuscript.
>
> ## Overclaiming Problem 7
> We appreciate your feedback on this point. Our original intent was to indicate that the parameter similarities across all layers are relatively high between base models and instruction-tuned models, suggesting that the base and instruction-tuned models share almost the same underlying knowledge.
>
> ## Missing Details Problem 1
> We apologize for the lack of clarity in describing how we derive the numeric values. Specifically, when applying the function MAXCOSSIM, we treat the elements in the returned vector as data points and use maximum likelihood estimation to compute the location parameter of the Gumbel distribution. This clarification has been incorporated into Section 3 in the revised manuscript.
>
> ## Missing Details Problem 2
> We appreciate the reviewer's suggestion. To address concerns regarding missing details, we have demonstrated how the DOCS method can distinguish between orthogonal matrices in the revised paper's Appendix A.5 (in the sense defined by Definition 3 in the paper).
>
> ## Missing Details Problem 3
> We apologize for the lack of clarity in describing how we derive the baselines for comparison. The baseline methods initially proposed were designed to measure representation similarity. However, since both representations and parameters are inherently matrices, it is technically feasible to adapt these baseline methods to compute parameter similarity. Therefore, in our implementation, we directly use the parameter matrices as the input for baseline methods.
>
> Notably, in LLM implementations, the rows of a weight matrix correspond to output dimensions, and the columns correspond to input dimensions. To align the column vectors with meaningful entities (e.g., neuron weights), similar to the DOCS method, we transpose the matrices $ W_v $, $ W_k $, $ W_q $, and $ \text{MLP-Up} $ before inputting them into the baselines.

---

> ### Author Response · Authors · 2024-11-23
>
> ## Need for Elaboration Problem 1
> We appreciate the reviewer's suggestions. We have already addressed these in the general reply and added the content to Appendix E.3 in the updated manuscript.
>
> As demonstrated in Figure 9 in Appendix E.3, our experimental results indicate a high degree of orthogonality among the weight matrices in LLMs. Similar observations have been documented in [1], which highlights the limitations of applying existing similarity indices directly to weight similarity in this domain due to their lack of distinguishability for orthogonal matrices.
>
> **Reference**
> [1] Tian, Y., Wang, Y., Zhang, Z., Chen, B., \& Du, S. (2023). Joma: Demystifying multilayer transformers via joint dynamics of MLP and attention. arXiv preprint arXiv:2310.00535.
>
> ## Need for Elaboration Problem 2
> We appreciate the reviewers for raising these suggestions.
> In Theorem 1, we establish the existence of a set of column-orthogonal matrices with significant differences, specifically:
>
> $ \| X - Y \|_F = \Omega(\sqrt{m}), $
>
> where the DOCS similarity is capable of reflecting these differences effectively:
>
> $ S_{\text{DOCS}}(X, Y) = \frac{1}{\sqrt{m}}. $
>
> In contrast, as shown in Table 1 of the paper, most existing methods yield a constant similarity value (independent of dimensionality) for any pair of orthogonal matrices, regardless of their differences. Theorem 1 theoretically demonstrates the superior discriminative power of the DOCS method for orthogonal matrices. Additionally, we provided a concrete example to illustrate this ability in distinguishing orthogonal matrices in Appendix A.5. These elaborations have been added to Section 3.1 of the revised manuscript for clarity.
>
> ## Question Problem 1
> Thank you for the suggestion. We have included these experimental results in Appendix F of the revised manuscript. In Figure 11 in Appendix F, we provide a visual comparison of eight different similarity indices applied to the MLP-UP layers of the reinitialized Meta-Llama-3.1-8B-Instruct model. We observed that all of them exhibit relatively random patterns.
>
> ## Question Problem 2
> We apologize for the confusion caused by our earlier wording. What we intended to convey is that a uniform configuration (at least during the SFT stage) does not fully utilize the inherent cluster structure of the model, and therefore, it may not be an optimal choice. We have revised the relevant statements in the updated manuscript for clarity.
>
> ## Question Problem 3
> Thank you for the suggestion. We have addressed this issue and revised the manuscript accordingly in the revised manuscript.

---

> > ### Comment · Reviewer_37uY · 2024-11-23
> > **Thanks for your detailed replies!**
> >
> > Thank you for your comprehensive replies and rewrite of the paper.
> >
> > I am mostly satisfied with your replies and edits of the paper, besides the proof in A.5.
> > This lemma doesn't prove that DOCS is discriminative on orthogonal matrices, but that there exist orthogonal matrices DOCS *can* discriminate.
> >
> >
> > Thanks again for revising the draft and responding to all my concerns and questions.
> > I will increase my scores accordingly.

---

> > > ### Author Response · Authors · 2024-11-24
> > >
> > > We sincerely appreciate the reviewer’s positive feedback on our response and are grateful for raising the scores!
> > >
> > > We apologize for any confusion caused by the expression in the lemma presented in Appendix A.5. To address this, we have revised the statement for clarity. In this lemma, we actually provide an explicit example demonstrating that there exist orthogonal matrices that DOCS can discriminate. According to Definition 3 of the paper, this illustrates the discriminative capability of DOCS for orthogonal matrices.

---

### Official Review · Reviewer_Fe2j · 2024-10-31

**Soundness:** 2
**Presentation:** 3
**Contribution:** 2
**Rating:** 5
**Confidence:** 4

**Summary:**

This paper presents a new metric for measuring the similarity between the weights of transformer models with the goal of gaining insights into LM learning. It is pointed out that existing methods like Canonical Correlation Analysis (CCA) and Centered Kernel Alignment (CKA) do not discriminate between orthogonal matrices well enough, which is a requirement for handling transformer weights. Moreover many existing metrics do not have certain desirable mathematical properties such as transformation and scaling invariance, symmetry, and reflexivity. The proposed metric, DOCS has all these properties.

Empirical justification for the suitability of DOCS for measuring similarity among transformer weights is provided by visualizing the similarity between pairs of MLP layers in a pretrained model.

The main findings using DOCS are that
1. neighboring layers in pretrained LMs are very similar to each other, and there exist clusters of similar layers, both indicating that the transformer architecture can be made more efficient by minimizing such redundancy.
2. the weights of base and instruction tuned models are generally highly similar to each other, and no clear trends exist at which layers weights are most dissimilar (i.e., impacted by instruction tuning).
3. some experts in MoE models are different from others.

**Strengths:**

The approach, mathematical notation, and the visualizations presented in the paper are easy to follow. The comparison of mathematical properties of similarity functions is informative.

**Weaknesses:**

**Motivation**: The main motivation behind studying the similarity between weights instead of representations is unclear. The arguments about discriminating between orthogonal matrices, and the mathematical properties of scaling invariance etc. are built on top of the assumption that studying similarity between the weight space is more insightful than in the representation space, but it is not clear to me why that is the case. After all, is it not the representations that directly affect the outputs and the behavior of a model? Moreover, representational similarity is arguably agnostic to the depth of a network, which is not the case with weight-space similarity. It would be great if the authors could clarify their motivation in the response.

**Empirical justification**: The empirical justification provided for DOCS being superior to other similarity measures is a) that the visualization of the similarity between layers lack a clear structure and appear noisy for CCA and Linear Regression in contrast to DOCS (Section 4.1), and b) the Gini coefficient of DOCS is higher than that of other metrics.
- (a) is not a convincing justification. A metric showing clearer patterns is not necessarily better.This could also mean that DOCS is missing out on useful information other metrics are capturing. It would be more meaningful to compare metrics against a ground truth, e.g., are models that have similar output distributions, like different training checkpoints of the same model, more similar according to one measure than the other?
- (b) requires elaboration. How is the Gini coefficient computed? Why is t relevant? How should one interpret these results?

**Impact of the findings**:
- That there is potential redundancy of knowledge in the weights of neural networks is an interesting finding, but it is not new. This has been empirically studied in many works that are well cited in this paper.
- The experiments comparing base and finetuned models and MoE do not show clear trends, and it is unclear what can be inferred from them.

**Questions:**

- It makes sense that the mathematical properties listed in Section 2 are desirable in a similarity measure of transformer weights. But are these properties *sufficient* for such a metric?

---

> ### Author Response · Authors · 2024-11-23
>
> ## Motivation
> Thank you for your comments regarding the motivation of our work. We have already addressed this in the general reply and added the content to Appendix E in the updated manuscript.
>
> ## Empirical Justification Problem 1
> We thank the reviewer for the insightful comment regarding the comparison of DOCS similarity indices over other methods. We have already addressed this in the general reply and added the content to Appendix E.3 in the updated manuscript.
>
> ## Empirical Justification Problem 2
> We sincerely thank the reviewer for the suggestions. The computation of the Gini coefficient in our work involves several steps. First, we exclude the diagonal elements from the similarity matrix as they are always equal to 1, focusing solely on inter-layer relationships. For each row, the remaining elements are normalized by dividing them by the sum of all row values, ensuring that the resulting coefficients reflect the relative distribution of similarity scores within the row. The Gini coefficient for each row is then calculated to quantify the inequality in the distribution of these scores, with higher values indicating a greater concentration of similarity among fewer layer pairs. Finally, the mean Gini coefficient across all rows is computed to summarize the overall inequality in the similarity matrix.
>
> A higher Gini coefficient suggests a more uneven distribution of similarity scores, implying a greater concentration of significant similarities in fewer layer pairs, which potentially reveals structural characteristics of the model parameters. As demonstrated in Table 2 of our paper, our proposed method (DOCS) achieves higher Gini coefficients compared to other methods, indicating its superior ability to isolate significant layer similarities. We have included clarifications and detailed explanations in the revised manuscript to address this point comprehensively.
>
> ## Impact of the Findings Problem 1
> We sincerely thank the reviewer for their valuable feedback and for recognizing the interest in exploring potential redundancies within neural network weights. While it is true that redundancy in neural networks has been previously studied, our work introduces a novel approach through the DOCS metric, offering fresh insights into this phenomenon. Specifically, DOCS enables a quantifiable and interpretable analysis of weight similarities, uncovering nuanced patterns that extend beyond prior findings. For example, we identified cluster structures among Transformer layers (Figure 4), which suggest deeper functional organization. Moreover, by analyzing similarities between base and instruction-tuned models, we demonstrated the strong preservation of foundational knowledge after fine-tuning, as well as distinct grouping tendencies of weight matrices ($W_q$ and $W_k$, $W_v$ and $W_o$), which may provide insights into optimization dynamics.
>
> Additionally, the application of DOCS to Mixture-of-Experts (MoE) architectures revealed clear distinctions among experts, highlighting their potential specialization. These contributions suggest that DOCS not only reinforces existing knowledge but also supports future innovations. For instance, the identified clusters can guide parameter-efficient techniques like LoRA by assigning adaptive ranks to clustered layers, aiding in model compression and structured knowledge distillation. We believe these aspects illustrate the distinct contributions of our method and its applicability to advancing neural network understanding and optimization.

---

> ### Author Response · Authors · 2024-11-23
>
> ## Impact of the Findings Problem 2
> We appreciate the reviewers' feedback and the opportunity to clarify our findings. Regarding the trends in the experiments comparing base and fine-tuned models and MoE, we have further updated and refined our analysis in the revised paper. We hope our analysis provides meaningful insights when interpreted in the context of the DOCS index and the architectural roles of the models.
>
> Specifically, for comparing base and fine-tuned models, DOCS scores exceeding $0.7$ consistently across all evaluated weight matrices indicate a strong preservation of foundational knowledge post-fine-tuning. Additionally, the variation patterns in the DOCS scores' curves allow weight matrices to be categorized into three distinct groups ($W_q$ and $W_k$, $W_v$ and $W_o$), potentially offering insights into the optimization dynamics structure during fine-tuning.
>
> For the MoE experiments, the visualization of expert weight matrices revealed clear distinctions among certain experts, as evidenced by the pronounced separation in similarity heatmaps. This phenomenon underscores the potential for individual experts to specialize in unique roles. We postulate that such behavior could be attributed to imbalances in data distribution during the training of the MoE architecture, wherein a significant portion of the input is directed toward one specific expert. These findings provide a basis for further investigation into the mechanisms underpinning parameter changes and expert specialization across model configurations [1],[2].
>
> **Reference**
> [1] Dai, D., Dong, L., Ma, S., Zheng, B., Sui, Z., Chang, B., \& Wei, F. (2022). Stablemoe: Stable routing strategy for mixture of experts. arXiv preprint arXiv:2204.08396.
> [2] Zuo, S., Liu, X., Jiao, J., Kim, Y. J., Hassan, H., Zhang, R., et al. (2021). Taming sparsely activated transformer with stochastic experts. arXiv preprint arXiv:2110.04260.
>
> ## Question
> We appreciate the reviewer's thoughtful feedback. The properties we have identified are intended to ensure that the similarity measure is meaningful. Specifically, these include:
>
> - **Permutation Transformation (PT) Invariance**: $S(X, Y) = S(XP\_X, YP\_Y)$ for any permutation matrices $P_X$ and $P_Y$.
> - **Symmetry**: $S(X, Y) = S(Y, X)$.
> - **Isotropic Scaling (IS) Invariance**: $S(aX, bY) = S(X, Y)$ for any non-zero scalars $a$, $b$.
> - **Reflexivity**: $S(X, X) = 1$.
> - **Discriminative on Orthogonal Matrices**.
>
> Together, these properties are sufficient because they collectively address both invariance to irrelevant transformations and sensitivity to differences in orthogonal matrices. By satisfying these criteria, the similarity measure can provide interpretable comparisons of transformer weights across various contexts.
>
> Further, other widely used similarity methods for models, such as those proposed by [1], [2], and [3], consider mathematical properties related to metrics that are subsets of the above properties.
>
> **Reference**
> [1] Kornblith, S., Norouzi, M., Lee, H., \& Hinton, G. (2019, May). Similarity of neural network representations revisited. In International conference on machine learning (pp. 3519-3529). PMLR.
>
> [2] Morcos, A., Raghu, M., \& Bengio, S. (2018). Insights on representational similarity in neural networks with canonical correlation. Advances in neural information processing systems, 31.
>
> [3] Raghu, M., Gilmer, J., Yosinski, J., \& Sohl-Dickstein, J. (2017). Svcca: Singular vector canonical correlation analysis for deep learning dynamics and interpretability. Advances in neural information processing systems, 30.

---

> ### Comment · Reviewer_Fe2j · 2024-11-25
>
> I thank the authors for their detailed response.
>
> ## Motivation
> I understand and agree with the authors argument that similarity in the weight space can reveal patterns that may not be evident from looking at similarity in the representation space. My question is about the *significance* of such patterns, particularly given that it is the representations that ultimately affect the outputs and the behavior of the model.
>
> For instance, in the example presented in Appendix E, the weight space analysis shows similarity between layers that is not evident in from the representation space analysis. But does this necessarily mean that there is redundancy in the model, given that the representations are actually dissimilar? Would it make sense to prune the model by removing layers with similar weights even when their output representations are not similar? I think it would make more sense to make pruning decisions based on representation space analyses.
>
> Are there any *actionable* insights that only weight space analyses can provide?
>
> ## Empirical justification from the Gini coefficient
> I thank the authors for the additional details about the computation of Gini coefficient and how to interpret it. I understand that this analysis highlights the differences between the similarity spaces, and the DOCS results in a more uneven distribution of similarity scores. However, I find it hard to interpret this result because there isn't a notion of ground truth here. If there were a true similarity metric, one could compare the Gini coefficient of that metric with DOCS and the other metrics, and see which one's coefficient is closes to the true metric. But without that, how does a greater unevenness in similarity scores necessarily mean a better metric?

---

> > ### Author Response · Authors · 2024-11-28
> >
> > ## Further Reply of Motivation
> > We sincerely thank the reviewer for the thoughtful feedback. We acknowledge the clear distinction the paper draws between representational similarity and weight similarity, and we realize the importance of emphasizing that both aspects are crucial. Representation and weight are two fundamental facets of a model, and a comprehensive understanding of LLMs requires consideration of both. While representation analysis offers profound insights, analyzing weight matrices further enriches our understanding of the structure and behavior of LLMs (see general reply (1) or Appendix E.1). We have updated the discussion of representation vs. weight in Section 1 to stress the significance of both.
> >
> > The insights revealed by DOCS regarding the model's weight structure have significant potential to guide applications. For instance, the identified cluster structures could be utilized to introduce inductive biases during the supervised fine-tuning stage of parameter-efficient techniques. One possible application involves leveraging clusters of highly similar weight layers to enable weight sharing or adaptive rank settings during fine-tuning. Such strategies may reduce computational costs and help preserve the base model's parameter structure despite supervised fine-tuning.
> >
> > Regarding the model pruning task scenario you proposed, existing structural pruning techniques typically consider the representation obtained before and after removing a certain layer; if the representations are highly similar, removing that layer is deemed feasible. We believe that combining model representation similarity with weight similarity information can lead to more effective strategies than relying solely on representation similarity. For instance, we can consider representation similarity while also accounting for the weight similarity between the layer to be removed and its adjacent layers. Priority can be given to removing layers where the representation remains highly similar after removal and the weight similarity with adjacent layers is high. Such layers may be more redundant.
> >
> > ## Further Reply of Empirical justification
> > We appreciate the reviewer's thoughtful feedback. As demonstrated in Figure 2, the similarity values produced by many other metrics often concentrate within a narrow range, resulting in minimal relative changes between different layers. For example, in subfigure (d) of Figure 2, the heatmap of SVCCA shows that the majority of similarity values between layers are clustered in the range of $0.64$–$0.70$. This concentration indicates that these metrics have limited power to capture subtle differences in the model's structure.
> >
> > To quantify these observations, we use the Gini coefficient, which measures the inequality among values in a distribution. A higher Gini coefficient for DOCS indicates a wider spread of similarity values, reflecting its enhanced ability to discriminate between layers. While we acknowledge the difficulty in defining a ground truth for such structural characteristics, we believe that the Gini coefficient provides a useful quantitative description of these differences.
> >
> > We hypothesize that the limited discriminative capability of other metrics may stem from their insensitivity to the similarity between orthogonal matrices, as discussed in Section 2 and Appendix E.2 of the manuscript. This insensitivity can cause them to overlook structural differences that DOCS can detect.
> >
> > To further illustrate the advantages of the DOCS method, we have included additional results in Appendix E.3 of the updated manuscript, as discussed in General Reply (3). These results provide further evidence from another perspective, demonstrating the advantages of DOCS in capturing the model's structural characteristics.

---

> > > ### Comment · Reviewer_Fe2j · 2024-11-29
> > >
> > > I thank the authors for additional explanations about the motivation and the use of Gini coefficient. I find it convincing that studying similarity in the weight space can provide insights that are complementary to those from the representation space. I am increasing my rating accordingly. I still believe that the practical utility of studying weight space similarity could be demonstrated by applying DOCS to cases where weight space similarity is beneficial, and this is within the scope of this paper.

---

> > > > ### Author Response · Authors · 2024-12-02
> > > >
> > > > We sincerely appreciate the reviewer’s positive feedback on our response and are grateful for raising the scores!

---

### Official Review · Reviewer_UcWi · 2024-11-03

**Soundness:** 3
**Presentation:** 2
**Contribution:** 3
**Rating:** 6
**Confidence:** 2

**Summary:**

This paper introduces DOCS, a novel matrix-similarity index that measures weight similarity of LLMs. It provides a more reliable and accurate measure of similarity between LLM weight matrices.

**Strengths:**

1.The approach is novel. It is a discriminative index which captures meaningful differences between weight matrices than other methods.

2.This paper encompasses detailed theoretical derivation on discriminative property and comprehensive experiments, which demonstrate discriminative nature of DOCS and provides some insights into the structure of LLMs.

**Weaknesses:**

1.Some of findings are not straightforward in this paper. For example, the authors associate multiple reasoning phases with clusters and attribute different numbers of clusters across LLMs to different training strategies. However, they do not clarify the rationale behind these conclusions.

2.Although the authors claim that DOCS satisfies all desired mathematical properties, they do not provide corresponding proofs in the paper.

**Questions:**

1.Could you provide a detailed explanation on the relation between clusters of similar phases and multiple reasoning phases?

2.Proofs of DOCS satisfying all desired mathematical properties could be added to improve the rigorousness of the claim.

3.I am curious about other similarity indices could achieve similar findings on LLMs. I believe this is important to further demonstrate the superiority of DOCS over other methods.

---

> ### Author Response · Authors · 2024-11-23
>
> ## Weaknesses 1 and Questions 1
> We appreciate your insightful comments. We have removed the discussion regarding the relationship between clusters within the LLM layers and reasoning phases from the manuscript, recognizing that it was speculative.
>
> However, we would like to clarify our original intent here. Consider an extreme case where the layers within each of the two clusters are identical. In this scenario, the first cluster functions by transforming the token embeddings into a different knowledge space—acting as an implicit encoder. The second cluster then transforms this encoded information back into the token embedding space, effectively decoding it to produce the next token. This leads us to conjecture that the first cluster corresponds to an encoding process that maps the input into a latent knowledge representation, while the second cluster decodes this representation to generate the output. We believe this interpretation offers a meaningful explanation for the structural patterns observed across different models.
>
> ## Weaknesses 2 and Questions 2
> Thank you for your constructive comments and suggestions. In response to the request for rigorous proofs of $DOCS$ satisfying the desired mathematical properties, we have revised our manuscript to include these proofs in Appendix A of the updated manuscript. These include proofs of Permutation Transformation Invariance, Symmetry, Isotropic Scaling Invariance, Reflexivity, and Discriminative on Orthogonal Matrices.
>
> ## Questions 3
> We thank the reviewer for the insightful comment regarding the comparison of similarity indices and the need to further demonstrate the advantages of DOCS over other methods. We have already addressed this in the general reply and added the content to Appendix E.3 in the updated manuscript.

---

> > ### Comment · Reviewer_UcWi · 2024-12-02
> >
> > Thank you for your detailed explanation for my questions. I especially appreciate the clarification of question 1 and additional information in Appendix E. I will maintain my positive score.

---

> > > ### Author Response · Authors · 2024-12-02
> > >
> > > We are sincerely grateful to the reviewer for the positive feedback of our response. We appreciate your support very much!

---

### Official Review · Reviewer_RK55 · 2024-11-04

**Soundness:** 3
**Presentation:** 3
**Contribution:** 3
**Rating:** 6
**Confidence:** 3

**Summary:**

The paper introduces the Distribution of Cosine Similarity (DOCS), a novel index designed to quantitatively assess the similarity between weight matrices in large language models (LLMs). This tool aims to enhance the analysis of LLM architectures by revealing significant patterns within them. Using DOCS, the authors identify that adjacent layers in contemporary open-source LLMs often show high weight similarity and tend to cluster, indicating a trend toward depth-wise functional specialization. Furthermore, the paper demonstrates that DOCS effectively quantifies similarity for orthogonal matrices, an important consideration given the common use of orthogonal initializations in LLMs.

**Strengths:**

1.	This paper is overall well-written.

2.	This paper proposes to better calculate the parameter similarity between different layers by computing the cosine similarity between corresponding vectors and analyzing their distribution.

3.	Using the technique they propose, the authors observed that: 1) neighboring transformer layers exhibit similar weights, 2) clusters of similar transformer layers exist, 3) the difference of parameters among different models.

**Weaknesses:**

1.	Some experimental designs could be more detailed and justified, such as the use of the gumbel distribution.

2.	No direct application could be derived from this analysis paper, which could be a weakness of the paper. The authors could add a discussion section to further point out the practical value of the findings in this paper.

**Questions:**

1.	Why did you choose to use Gumbel distribution to summarize each distribution to a parameter? Is there any specific reason or proof for that?

2.	Could you suggest some other potential applications besides what people have already done (e.g., leveraging layer clusters to reduce computational requirements) that could benefit from the findings in this paper?

---

> ### Author Response · Authors · 2024-11-23
>
> ## Weaknesses 1 and Questions 1
>
> We appreciate the reviewer's insightful question regarding our choice of the Gumbel distribution to summarize the maxima of the cosine similarities. To address your concern, we provide a more detailed mathematical justification grounded in Extreme Value Theory (EVT).
>
> In our algorithm, for each column $A\_j$, we compute the maximum absolute cosine similarity:
> $$
> s\_{A\_j} = \max\_{k} |C\_{jk}|, \quad \text{where} \quad C\_{jk} = \frac{A\_j^\top B\_k}{\|A\_j\| \|B\_k\|}.
> $$
> This results in a set of maxima $\mathbf{s}\_X = [s\_{A\_1}, s\_{A\_2}, \dots, s\_{A\_m}]^\top$.
>
> According to the Fisher-Tippett-Gnedenko theorem, the limiting distribution of the maxima of a sequence of independent and identically distributed (i.i.d.) random variables, after appropriate normalization, converges to one of three types: Gumbel, Fréchet, or Weibull. Theoretical results and empirical studies [1] have shown that for bounded variables, the Gumbel distribution can provide a good approximation for the distribution of maxima.
>
> By fitting a Gumbel distribution to $\mathbf{s}\_X$ and $\mathbf{s}\_Y$, we obtain the location parameters $u\_X$ and $u\_Y$, which summarize the central tendency of the extreme similarities between $X$ and $Y$. The similarity index $S\_{\text{DOCS}}$ is then computed as:
> $$
> S_{\text{DOCS}} = \frac{u\_X + u\_Y}{2}.
> $$
>
> **Reference**
> [1] Leadbetter, M. R., Lindgren, G., & Rootzén, H. (2012). Extremes and related properties of random sequences and processes. Springer Science & Business Media.
>
> ## Weaknesses 2 and Questions 2
>
> We appreciate the reviewer's constructive feedback and agree that discussing practical applications would strengthen the impact of our paper. Below, we address the comment by suggesting several additional potential applications besides what people have already done (e.g., leveraging layer clusters to reduce computational requirements) of our findings:
>
> 1. **Incorporating Inductive Biases During the Supervised Fine-Tuning Stage:**
>    Building on the identified cluster structures, we propose introducing specific inductive biases in the fine-tuning stage of parameter-efficient fine-tuning methods, such as LoRA. For example, we can set adaptive ranks for layers within the cluster to minimize disruptions to the base model's original parameter structure.
>
> 2. **Guiding Sparsity Patterns for Model Compression:**
>    The clustering results could inform sparsity patterns by selectively pruning redundant connections within clustered layers. Such targeted sparsification could potentially reduce model size and computational costs while preserving performance, especially in resource-constrained deployment scenarios.
>
> 3. **Enhancing Knowledge Distillation Strategies:**
>    The layer clustering could also be used to identify critical layers that should be prioritized during the knowledge distillation process. For instance, a student model could focus on mimicking the most representative layers within each cluster, reducing the need to replicate the entire teacher model's architecture.
>
> We have incorporated these potential future work ideas into section 5 in the revised manuscript to illustrate the broader implications and potential applications of our findings.

---

> > ### Comment · Reviewer_RK55 · 2024-11-26
> >
> > Thanks you a lot for your reply, I'll maintain my score.

---

> > > ### Author Response · Authors · 2024-11-28
> > >
> > > We are sincerely grateful to the reviewer for the positive feedback of our response. We appreciate your support very much!

---

### Official Review · Reviewer_6MYG · 2024-11-04

**Soundness:** 3
**Presentation:** 4
**Contribution:** 4
**Rating:** 8
**Confidence:** 5

**Summary:**

The paper introduces Distribution of Cosine Similarity (DOCS), a new method for measuring similarity between LLM weights.  Unlike previous methods that focused on analyzing representations, DOCS directly examines weight matrices, providing a more precise tool for understanding the internal structure of these complex models. DOCS reveales two important insights, (i) adjacent layers in LLMs often share similar weights and (ii) tend to form functional clusters suggesting depth-wise functional specialization. The findings show that the conventional use of uniform layer configurations in large language models may not be optimal, suggesting hints for more efficient architectures design.

**Strengths:**

- The method provides a quantitatively similarity score for LLM weights (not representation) which may be helpful for model merging and pruning research.
- The method is proposed with a good level of interpretability and visualization with impressive analysis.

**Weaknesses:**

While the paper showed an impressive set of analysis (with page limitations), I would very much like to see an analysis on what makes DOCS work. Especially in Algorithm 1, which step is the most crucial one. A small set of experiments may help.

**Questions:**

- Please follow weakness

---

> ### Author Response · Authors · 2024-11-23
> **Weaknesses 1**
>
> We greatly appreciate the reviewer’s insightful comment regarding the need for further analysis on what makes the DOCS algorithm effective. In response, we have added some additional experimental results in Appendix G of the updated manuscript.
>
> We illustrate the intermediate computations for the $MLP\text{-}Up$ parameter matrices from the 4th and 8th blocks of the Meta-Llama-3.1-8B model, presenting frequency distribution histograms of the resulting similarity vectors $\mathbf{s}_X$ and $\mathbf{s}_Y$ (Figure 12 in Appendix G). These results demonstrate the intermediate outcomes of the DOCS method.
>
> Furthermore, we conducted an analysis comparing the vanilla DOCS algorithm with an average-based variant. As shown in Figure 13 in Appendix G, the average-based DOCS approach produces heatmaps that fail to capture meaningful structural patterns. This analysis demonstrates that the max operation in MaxCosSim is crucial for extracting salient features.

---

> > ### Comment · Reviewer_6MYG · 2024-11-27
> >
> > Thank you for your reply. I really appreciate the reply, specially the new section G. Considering the entire contribution of the work, I've decided to increase my score.

---

> > > ### Author Response · Authors · 2024-11-28
> > >
> > > We sincerely appreciate the reviewer’s positive feedback on our response and are grateful for raising the scores!

---

### Author Response · Authors · 2024-11-23
**Reference**

[1] Morcos, A., Raghu, M., & Bengio, S. (2018). Insights on representational similarity in neural networks with canonical correlation. Advances in neural information processing systems, 31.

[2] Raghu, M., Gilmer, J., Yosinski, J., & Sohl-Dickstein, J. (2017). Svcca: Singular vector canonical correlation analysis for deep learning dynamics and interpretability. Advances in neural information processing systems, 30.

[3] Kornblith, S., Norouzi, M., Lee, H., & Hinton, G. (2019, May). Similarity of neural network representations revisited. In International conference on machine learning (pp. 3519-3529). PMLR.

[4] Tian, Y., Wang, Y., Zhang, Z., Chen, B., & Du, S. (2023). Joma: Demystifying multilayer transformers via joint dynamics of mlp and attention. arXiv preprint arXiv:2310.00535.

---

### Author Response · Authors · 2024-11-23
**General Reply (3): Advantages of DOCS Compared to Other Similarity Indices**

In addition to the discussions on the advantages of DOCS compared to other similarity indices presented in Section 4.1 of the paper, we have included *additional experimental results* in Appendix E.3 of the updated manuscript.

In Section 4.1, Figure 2 showcases the evaluation outcomes of eight different similarity measures on the $MLP\text{-}Up$ layers from the Meta-Llama-3.1-8B-Instruct model. The resulting heatmaps highlight that metrics like Linear Regression, Canonical Correlation Analysis (CCA), and CCA (Nuclear) fail to display discernible structural patterns. On the other hand, methods such as Singular Vector CCA (SVCCA), SVCCA (Nuclear), and Linear Centered Kernel Alignment (Linear CKA) exhibit faint block-like or striped formations in the off-diagonal regions, which might be attributed to either noise or inherent limitations in the measures themselves. These anomalies could arise due to the reduced sensitivity of these indices in differentiating between orthogonal matrices, as elaborated in Section 2.

In Appendix E.3 of the updated manuscript, to better illustrate the superiority of the DOCS method compared to other approaches, we conducted *further* experiments using three models:

- (A) `meta-llama/Meta-Llama-3.1-8B` (the base model)
- (B) `meta-llama/Meta-Llama-3.1-8B-Instruct` (an instruction-tuned version of the base model)
- (C) A version of `meta-llama/Meta-Llama-3.1-8B` with randomly initialized weights

Intuitively, the weight matrices of corresponding layers in models (A) and (B) should exhibit significantly higher similarity compared to those between models (A) and (C), since model (C) contains random weights and lacks the learned structure present in models (A) and (B).

To quantify this, we define the *similarity ratio* as the ratio of the similarity scores between models (A) and (B) to those between models (A) and (C). Specifically, for each similarity index, we compute the similarity of the corresponding $MLP\text{-}Up$ and $MLP\text{-}Down$ weight matrices between models (A) and (B), and between models (A) and (C). A higher ratio indicates that the similarity index is better at distinguishing between meaningful relationships in model weights (as seen in models (A) and (B)) and unrelated weights (as seen in models (A) and (C)). Thus, a higher ratio reflects the ability of the index to highlight structural patterns specific to related models while minimizing noise from uncorrelated data.

As shown in Figure 10, the experimental results reveal that the similarity ratio computed using the DOCS method is much higher—approximately 10 times greater—than those obtained with the other indices. This demonstrates the effectiveness of DOCS in capturing meaningful similarity patterns in model weights.

---

### Author Response · Authors · 2024-11-23
**General Reply (2) : Motivation of DOCS**

### Non-Discriminative for Orthogonal Matrices
In Section 1, we emphasized that many existing similarity indices, such as Canonical Correlation Analysis (CCA) [1], Singular Vector Canonical Correlation Analysis (SVCCA) [2], and linear Centered Kernel Alignment (linear CKA) [3], are *non-discriminative for orthogonal matrices*. An *orthogonal matrix* $Q$ is defined by the property $Q^\top Q = Q Q^\top = I$, where $I$ is the identity matrix. This non-discriminative nature means that these indices can yield the same score when assessing the similarity between *any two* orthogonal matrices, regardless of their actual differences. These conclusions are uniformly shown in Table 2, with proofs provided in Appendix B. This issue is *particularly relevant* in the context of LLMs, where orthogonal matrices commonly occur throughout the training process [4].

In fact, in Figure 2, we observe that when CKA is directly used to measure weight similarity, a large number of values concentrate in the high range of 0.78–0.80. This is likely due to the non-discriminative property for orthogonal matrices.

In Appendix E.2 of the updated manuscript, we provide a *further result* indicating a high degree of orthogonality among the weight matrices in LLMs.
To quantify this orthogonality, we design an index named the Off-diagonal Average Cosine Similarity metric. This metric provides a systematic way to measure the extent of orthogonality between columns of a given weight matrix. The formulation of the metric is as follows:

$$
\text{Off-Diagonal Average Cosine Similarity} (\mathbf{X}) = \frac{1}{n(n-1)} \sum_{i=1}^{n} \sum_{\substack{j=1 \\ j \neq i}}^{n} \left| \frac{\mathbf{x}_i \cdot \mathbf{x}_j}{\|\mathbf{x}_i\| \, \|\mathbf{x}_j\|} \right|,
$$

where $\mathbf{X}$ is an $n$-column matrix, and $\mathbf{x}_i$ represents the $i$-th column of $\mathbf{X}$. From this formulation, it can be observed that the larger the Off-Diagonal Average Cosine Similarity value, the smaller the orthogonality of the matrix.

We constructed a family of approximately orthogonal matrices, denoted as $\mathcal{M}\_\theta$, which are defined as follows:

$$
\mathcal{M}\_\theta = I\_{n} + \theta  v  \mathbf{1}^\top
$$

where:
- $I_{n}$ represents the identity matrix of size $n \times n$.
- $\theta \in \mathbb{R}$ is a scalar that controls the magnitude of orthogonality; smaller values of $\theta$ correspond to matrices that are closer to orthogonal.
- $v \in \mathbb{R}^{n}$ is a random vector sampled from a normal distribution, specifically $v \sim \mathcal{N}(0, 1)$.
- $\mathbf{1} \in \mathbb{R}^{n}$ is a vector of ones.

We utilized the Q matrices and O matrices from each layer of the Meta-Llama-3.1-8B-Instruct model. Additionally, we generated four sets of approximately orthogonal matrices by sampling $\mathcal{M}\_\theta$ with $\theta$ values of 0.005, 0.003, 0.002, and 0.001, ensuring that their shapes matched those of the Q and O matrices. We then computed the Off-Diagonal Average Cosine Similarity for these matrices.

The experimental results, presented in Figure 9, reveal that the majority of the Q and O matrices from each layer of the Meta-Llama-3.1-8B-Instruct model exhibit stronger orthogonality compared to $\mathcal{M}\_{0.003}$. According to the definition of $\mathcal{M}\_\theta$, $\mathcal{M}\_{0.003}$ only has a very small perturbation added to the identity matrix. This observation suggests that the Q and O matrices in the Meta-Llama-3.1-8B-Instruct model are highly orthogonal.

---

### Author Response · Authors · 2024-11-23
**General Reply (1): Motivation of DOCS**

As introduced in Section 1, while prior research has explored many methods for characterizing the similarity of neural networks --- such as Canonical Correlation Analysis (CCA) [1], Singular Vector Canonical Correlation Analysis (SVCCA) [2], and linear Centered Kernel Alignment (linear CKA) [3] --- applying these methods to weight matrices presents challenges due to two key factors: **Focus on Representation, Not Weights** and **Non-Discriminative for Orthogonal Matrices**. In addition to the discussions in Section 1 of the manuscript, we have added additional experimental results in Appendix E.1 and E.2 of the updated manuscript.

### Focus on Representation, Not Weights
In Section 1, we emphasized that representation and weight are two aspects of the model. Similar representations across layers do not necessarily imply similar weight matrices. This discrepancy arises from the use of *residual connections* in transformer architectures. This is evidenced by Figures 1a and 1b, which show that the input and output of the feedforward network have similar patterns of representation similarity.

In Appendix E.1 of the updated manuscript, we provide a *further result* showing that weight similarity helps discover intriguing patterns in open-source LLMs that would otherwise be overlooked. We conducted experiments on the `01-ai/Yi-1.5-9B-Chat` model. On one hand, we used the Linear CKA method to calculate the similarity of the $MLP\text{-}Up$ layer outputs in each block of the model. On the other hand, we used the DOCS method to calculate the similarity of the $MLP\text{-}Up$ parameters in each block. The experimental results are shown in Figure 8.

The results in Figure 8 underscore the strength of weight similarity compared to representational similarity methods in revealing structural patterns within the model. Specifically, the weight similarity shown in Subfigure 8a highlights a notable correspondence between layers 9–24 and layers 25–40. This suggests that the model may have employed a training strategy that duplicates a section of layers (9–24) to increase the model's size while saving on training costs.

Our analysis provides tools that can help reveal such structures in open-source LLMs. In contrast, Subfigures 8b, 8c, and 8d present the representational similarity of the $MLP\text{-}Up$ outputs. Despite using multiple input sentences for the analysis, these results primarily provide a limited understanding of weight structure. They exhibit relatively homogeneous patterns, lacking the fine-grained layer correspondence seen with weight similarity analysis. Therefore, in this example, weight similarity helps uncover patterns that would otherwise remain hidden, deepening our understanding of the model.

---

### Meta-Review · Area_Chair_Q3yi · 2024-12-19

**Metareview:**

This paper introduces DOCS (Distribution of Cosine Similarity), a novel similarity index for measuring weight similarity in LLMs. It directly analyzes weight matrices for discovering of structural patterns in LLMs,  such as layer redundancy, functional specialization, and weight preservation across fine-tuning. DOCS demonstrates strong performance over established similarity indices (e.g., CCA, SVCCA) and satisfies desirable mathematical properties like scaling invariance, symmetry, and orthogonal matrix discrimination.

Strengths:
- The proposed approach is a novel and computationally efficient, in addition to provides insights into LLMs' internal structures.
- It is well-grounded in theory, addressing key issues like orthogonal matrix discrimination.

Weakness:
- Reviewers raised concerns about the evaluation of DOCS, particularly its superiority over other indices without clear ground truth. While the authors introduced metrics like the Gini coefficient to quantify performance, its interpretability remains subjective.
- In addition, after rebuttal, some claims still remain speculative and lacked empirical support, such as the relationship between layer clusters and reasoning phases.

**Additional Comments On Reviewer Discussion:**

Reviewers reached a consensus to accept the paper, despite concerns about insufficient justification for DOCS’s advantages and overclaiming certain findings. The authors addressed these points during the rebuttal:

- The authors clarified that DOCS captures distinct patterns in similarity distributions (quantified by Gini coefficients). While no absolute ground truth was established, reviewers acknowledged the value of these insights.
- The authors mentioned that speculative claims would be revised or removed, improving the manuscript's focus. I found this is okay as the these speculative claims are not critical for the papers contributions.

---

### Decision · Program_Chairs · 2025-01-22

Accept (Poster)